# A comparison study on jacket substructures for offshore wind turbines based on optimization

Jan Häfele[1], Cristian G. Gebhardt[1], and Raimund Rolfes[1]

[1]Leibniz Universität Hannover/ForWind, Institute of Structural Analysis, Appelstr. 9a, 30167 Hannover, Germany

*Correspondence to:* Jan Häfele (j.haefele@isd.uni-hannover.de)

**Abstract.** The structural optimization problem of jacket substructures for offshore wind turbines is commonly considered as a pure tube dimensioning problem, minimizing the entire mass of the structure. However, this approach goes along with the assumption that the given topology is fixed in any case. The present work contributes to the improvement of the state of the art by utilizing more detailed models for geometry, costs, and structural design code checks. They are assembled in an optimization scheme, in order to consider the jacket optimization problem from a different point of view that is closer to practical applications. The conventional mass objective function is replaced by a sum of various terms related to the cost of the structure. To address the issue of high demand of numerical capacity, a machine learning approach based on Gaussian process regression is applied to reduce numerical expenses and enhance the number of considered design load cases. The proposed approach is meant to provide decision guidance in the first phase of wind farm planning. A numerical example for a NREL $5\,\mathrm{MW}$ turbine under FINO3 environmental conditions is computed by two effective optimization methods (sequential quadratic programming and an interior-point method), allowing for the estimation of characteristic design variables of a jacket substructure. In order to resolve the mixed-integer problem formulation, multiple subproblems with fixed integer design variables are solved. The results show that three-legged jackets may be preferable to four-legged ones under the boundaries of this study. In addition, it is shown that mass-dependent cost functions can be easily improved by just considering the number of jacket legs to yield more reliable results.

## 1 Introduction

The substructure contributes significantly to the total capital expenses of offshore wind turbines and thus to the levelized costs of offshore wind energy, which are still high compared to the onshore counterpart (Mone et al., 2017). Cost breakdowns show ratios of about $20\,\%$ (such as The Crown Estate, 2012; BVGassociates, 2013) depending on rated power, water depth, and what is considered as capital expenses. In the face of wind farms with often more than 100 turbines, it is easily conceivable that a slight cost reduction can render already substantial economical advantages to prospective projects. Structural optimization is paramount, because it provides the great opportunity to tap cost saving potential with low economical effort. Technologically, it is expected that the jacket will supersede the monopile when reaching the imminent turbine generation or wind farm locations with intermediate water depths from about $40$ to $60\,\mathrm{m}$ (see for instance Seidel, 2007; Damiani et al., 2016). According to current studies, there is an increasing market share of jackets (Smith et al., 2015). As it allows for many variants of structural

design, the jacket structure is therefore a meaningful object of structural optimization approaches, which benefits massively from innovative design methods and tools (van Kuik et al., 2016).

State of the art in the field of jacket optimization[1] is to deal with optimal design in terms of a tube dimensioning problem, where the topology is fixed. Structural design codes require the computation of time domain simulations to perform structural code checks for fatigue and ultimate limit state. As environmental conditions in offshore wind farm locations vary strongly, commonly thousands of simulations are necessary to cover the effect of varying wind and wave states for verification[2]. Therefore, numerical limitations are a great issue in state-of-the-art jacket optimization approaches. In literature, different approaches were presented to address this issue. Schafhirt et al. (2014) proposed an optimization scheme based on a meta-heuristic genetic algorithm to guarantee global convergence. To increase the numerical efficiency, a reanalysis technique was applied. Later, an improved approach was illustrated (Schafhirt et al., 2016), where the load calculation was decoupled from the actual tube dimensioning procedure and a simplified fatigue load set (Zwick and Muskulus, 2016) was applied. Similar approaches by Chew et al. (2015, 2016) and Oest et al. (2016) applied sequential quadratic or linear programming methods, respectively, with analytically derived gradients. Other optimization approaches using meta-heuristic algorithms were reported by AlHamaydeh et al. (2017) and Kaveh and Sabeti (2018), however, without comprehensive load assumptions. The problem of discrete design variables was addressed by Stolpe and Sandal (2018). Oest et al. (2018) presented a jacket optimization study, where different simulation codes were deployed to perform structural code checks. All mentioned works, except for the last one, represent tube sizing algorithms applied to the OC4 jacket substructure[3] (Popko et al., 2014) for the National Renewable Energy Laboratory (NREL) $5\,\mathrm{MW}$ reference turbine (Jonkman et al., 2009), where the initial structural topology is maintained even in case of strong tube diameter and wall thickness variations. Furthermore, it can be stated that all proposals share the entire mass of the jacket as objective function to be minimized, which is meaningful in terms of tube sizing. Due to numerical limitations, the utilized load sets are altogether small, for instance with low numbers of production load cases or the omission of special extreme load events. These assumptions constitute drawbacks when considering jacket optimization as part of a decision process in early design stages, where basic properties like the numbers of legs or bays are more critical than the exact dimensions of each single tube. Therefore, an optimization scheme, which addresses the early design phase, is highly desirable to provide decision guidance for experienced designers. Proposals tackling this kind of problem were given by Damiani (2016) and Häfele and Rolfes (2016), where technically oriented jacket models were proposed, however, lacking fatigue limit state checks in the first and detailed load assumptions in the second case. Based on the latter and with improved load assumptions, a hybrid jacket for offshore wind turbines with high rated power was designed (Häfele et al., 2016). Due to innovative materials (the technology readiness level of such a structure is still low), this work lacked detailed cost assumptions. Another proposal for an integrated design approach was made by Sandal et al. (2018), considering varying bottom widths and soil properties. This

---

[1]This work focuses on the problem of jacket optimization and disregards other substructure types. For a comprehensive overview of the structural optimization of wind turbine support structures, it is referred to Muskulus and Schafhirt (2014).

[2]During conceptual design phases, the number of load cases is commonly reduced.

[3]It is worth mentioning that the Offshore Code Comparison Collaboration Continuation (OC4) jacket is actually a structurally reduced derivation of the so-called UpWind jacket (Vemula et al., 2010), which was created to ease calculations within the verification efforts in the OC4 project. Therefore, it is not guaranteed that the OC4 jacket is an appropriate comparison object, as it does not incorporate details of tubular joints.

work is meant as an approach for conceptual design phases. However, concluding the state of the art, an optimization approach without massive limitations is still missing.

This work is intended as a contribution to the improvement of the state of the art by considering jacket optimization in a different way. Compared to other works in this field, the focus is on:

1. The incorporation of topological design variables in the optimization problem, while the dimensioning of tubes is characterized by global design variables,

2. More detailed cost assumptions,

3. More comprehensive load sets for fatigue and ultimate limit state structural design code checks,

4. A change in the exploitation of jacket optimization results. This work intends to consider jacket optimization as a part of the preliminary design phase, because it is assumed that the (economically) most expensive mistakes in jacket design are made in this stage of the design process.

A basis to address these points was elaborated by Häfele et al. (2018a), where appropriate geometry, cost, and structural code check models for fatigue and ultimate limit states were developed. In this study, these models are deployed within an optimization scheme to obtain optimal design solutions for jacket substructures. A more efficient or accurate method to solve the optimization problem is deliberately not provided in this study. The authors believe that there are numerous techniques presented in literature that are able to solve the jacket optimization problem.

The paper is structured as follows: Sect. 2 describes the technical and mathematical problem statements. Both the objective and the constraints are presented and explained in sect. 3. The optimization approach and methods to solve the problem are discussed in sect. 4. Sect. 5 illustrates the application of the approach to a test problem, a comparison of jackets with different topologies, performed for a NREL 5 MW turbine under FINO3 environmental conditions. This section comprises a detailed setup of the problem and a discussion of the results. The work is finalized with a consideration of benefits and limitations (sect. 6) and conclusions (sect. 7).

## 2   Problem statement

This paper presents a study on jacket substructures, based on optimization. The design of jackets is a complex task that requires profound expertise and experience. Therefore, it has to be clarified that this work does not provide a method replacing established design procedures. It is rather meant as a guidance in early design phases, where it is desirable to define the basic topology and dimensions of the substructure. In industrial applications, this step is commonly highly dependent on the knowledge of experienced designers. Along with this statement, it has to be pointed out that the term "optimal solution" may indicate a solution that it is indeed optimal concerning the present problem formulation, but not necessarily optimal in terms of a final design, which arises from the following aspects:

- Although the approach deploys more detailed assumptions on the modeling of costs and environmental conditions, compared to optimization approaches known from literature, it still incorporates simplifications, mainly for the sake of numerical efficiency.

- No sizing of each single tube is performed, for the same reason. This is a matter of subsequent design phases and tube dimensioning approaches exist in literature. Instead, tube dimensions are derived by global design variables.

- The design of pile foundation and transition piece is not performed in this approach. The reason is that both are considered in models of the structure and the costs, but are not impacted by the selected design variables.

- Only fatigue and ultimate limit state are assumed to be design-driving design constraints. Serviceability limit state, i.e., eigenfrequency constraints, is not considered as design-driving in this work, because the modal behavior of a wind turbine with jacket substructure is strongly dominated by the relatively soft tubular tower. In addition, a design leading to eigenfrequencies close to 1P- or 3P-excitation would probably fail due to high fatigue damages. Although the modal behavior is also impacted by the foundation, this is not significant here, as no foundation design is performed.

The overall goal of jacket optimization can be interpreted as a cost minimization problem involving certain design constraints. As stated before, it is assumed that the design-driving constraints of jackets are fatigue and extreme loads. In other words, a set of design variables for a parameterizable structure that minimizes its costs, $C_{total}$, is desirable, while fatigue and ultimate limit state constraints are satisfied. I.e., the maximal normalized[4] tubular joint fatigue damage (among all tubular joints), $h_{FLS}$, is less than or equal to 1 and the extreme load utilization ratio (among all tubes), $h_{ULS}$, is less than or equal to 1.

The total expenses are defined as objective function $f(\boldsymbol{x})$, which depends on an array of design variables, $\boldsymbol{x}$:

$$f(\boldsymbol{x}) = \log_{10}\left(C_{total}(\boldsymbol{x})\right). \tag{1}$$

In this equation, the cost value is logarithmized to obviate numerical issues. The constraints, $h_1(\boldsymbol{x})$ and $h_2(\boldsymbol{x})$, are formulated so as to match the requirements of mathematical problem statements, thus:

$$h_1(\boldsymbol{x}) = h_{FLS}(\boldsymbol{x}) - 1,$$
$$h_2(\boldsymbol{x}) = h_{ULS}(\boldsymbol{x}) - 1, \tag{2}$$

depending also on the array of design variables, $\boldsymbol{x}$.

Based on the technical problem statement, we define the mathematical problem statement in terms of a nonlinear program:

$$\begin{aligned} \min \quad & f(\boldsymbol{x}) \\ \text{such that} \quad & \boldsymbol{x}_{lb} \leq \boldsymbol{x} \leq \boldsymbol{x}_{ub}, \\ & h_1(\boldsymbol{x}) \leq 0 \text{ and } h_2(\boldsymbol{x}) \leq 0, \end{aligned} \tag{3}$$

---

[4]All fatigue damages are normalized in the way that the lifetime fatigue damage corresponds to a value of 1.

where $x$ is the array or vector of design variables, $x_{lb}$ and $x_{ub}$ are the lower and upper boundaries, respectively, $f(x)$ is the objective function, covering the costs related only to the substructure, and $h_1(x)$ and $h_2(x)$ are nonlinear constraints representing structural code checks for fatigue and ultimate limit state that are required to be satisfied for every design.

## 3 Objective and constraints

This section illustrates the jacket model, which is the basis for the optimization study. Moreover, the models for costs and structural design code checks are described, which depict the objective and constraint functions, respectively. These models were elaborated in a previous work (Häfele et al., 2018a).

### 3.1 Jacket modeling and design variables

In this work, it is assumed that a jacket substructure can be described by 20 parameters in total, from which ten define topology,
seven tube dimensions, and three material properties. Topological parameters are the number of legs, $N_L$, number of bays, $N_X$ (both integer variables), foot radius, $R_{foot}$, head-to-foot radius ratio, $\xi$, jacket length, $L$, elevation of the transition piece over mean sea level, $L_{MSL}$, lowermost segment height, $L_{OSG}$, uppermost segment height, $L_{TP}$, the ratio of two consecutive bay heights, $q$, and a boolean flag, $x_{MB}$, determining whether the jacket has mud braces (horizontal tubes below the lowermost layer of K joints) or not. The topology of one example with four legs ($N_L = 4$), four bays ($N_X = 4$), and mud braces $x_{MB} = \text{true}$
is shown in Figure 1. The tube sizing parameters are the leg diameter, $D_L$, and six dependent parameters defining relations between tube diameters and wall thicknesses at the bottom and top of the structure: $\gamma_b$ and $\gamma_t$ are the leg radius-to-thickness ratios, $\beta_b$ and $\beta_t$ are the brace-to-leg diameter ratios, and $\tau_b$ and $\tau_t$ are the brace-to-leg thickness ratios, where the indices, $b$ and $t$, indicate values at the bottom and the top of the jacket, respectively. Using dependent parameters is beneficial, because structural code checks are valid for certain ranges of these dependent variables. Furthermore, for structural analysis, the material
is assumed to be isotropic and can thus be described by a Young's modulus, $E$, a shear modulus, $G$, and density, $\rho$.

To decrease the dimension of the problem, height measures related to the location of the wind farm ($L$, $L_{MSL}$, $L_{OSG}$, $L_{TP}$) and the material parameters ($E$, $G$, $\rho$) are fixed. In addition, it is supposed that each design has mud braces ($x_{MB} = \text{true}$). Although designs without mud braces are also imaginable, fixing this parameter is advantageous, as it is not continuous. The array of design variables has therefore a dimension of 12:

$$x = (N_L\ N_X\ R_{foot}\ \xi\ q\ D_L\ \gamma_b\ \gamma_t\ \beta_b\ \beta_t\ \tau_b\ \tau_t)^T. \qquad (4)$$

The number of design variables is not necessarily minimal, but on the one hand mathematically manageable and on the other hand meaningful from the technical point of view.

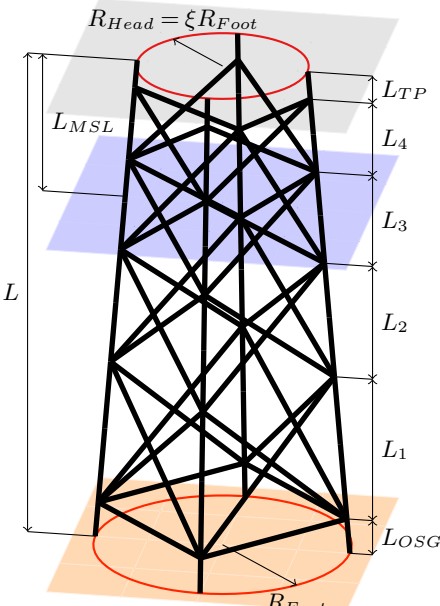

**Figure 1.** Jacket geometry model with variables characterizing the topology of the structure, shown exemplarily for a jacket with four legs, four bays, and mud braces. The ground layer is illustrated by ▬, the mean sea level and transition piece layers by ▬ and ▬, respectively.

## 3.2 Cost function (objective)

The total capital expenses, $C_{total}$, comprise several terms, $C_j$, expressed as sum of so-called factors, $c_j$, weighted by unit costs[5], $a_j$:

$$C_{total}(\boldsymbol{x}) = \sum C_j(\boldsymbol{x}) = \sum a_j c_j(\boldsymbol{x}). \tag{5}$$

A factor may be any property of the structure describing a cost contribution that can be expressed in terms of the design variables. A pure mass-dependent cost modeling approach, as used in most optimization approaches, would involve only one factor, while no unit cost value is required for weighting. However, a realistic cost assessment involves more than only the structural mass. For example, in case of a structure with very lightweight tubes, but many bays, it can be imagined that the manufacturing costs tend to be a cost-driving factor. To consider known, important impacts on jacket capital expenses, seven

---

[5]Unit cost values are given in section 5.3.

factors are incorporated. Namely, expenses for material, $C_1$, depending on the mass, $c_1$:

$$
\begin{aligned}
c_1(\boldsymbol{x}) =& 2\rho N_L \pi D_L^2 \sum_{i=1}^{N_X} \left( \left( \frac{\beta_i \tau_i}{2\gamma_i} + \frac{\tau_i^2}{4\gamma_i^2} \right) \sqrt{\frac{L_i^2}{\cos^2(\Phi_p)} + (R_i + R_{i+1})^2 \sin^2\left(\frac{\vartheta}{2}\right)} \right) \\
&+ x_{MB} \rho N_L \pi D_L^2 \left( \frac{\beta_b \tau_b}{2\gamma_b} + \frac{\tau_b^2}{4\gamma_b^2} \right) 2R_1 \sin\left(\frac{\vartheta}{2}\right) \\
&+ \rho N_L \pi D_L^2 \sum_{i=1}^{N_X} \left( \left( \frac{1}{2\gamma_i} + \frac{1}{4\gamma_i^2} \right) \frac{L_{m,i}}{\cos(\Phi_s)} + \left( \frac{1}{2\gamma_{i+1}} + \frac{1}{4\gamma_{i+1}^2} \right) \frac{(L_i - L_{m,i})}{\cos(\Phi_s)} \right) \\
&+ \rho N_L \pi D_L^2 \left( \frac{1}{2\gamma_b} + \frac{1}{4\gamma_b^2} \right) \frac{L_{OSG}}{\cos(\Phi_s)} \\
&+ \rho N_L \pi D_L^2 \left( \frac{1}{2\gamma_t} + \frac{1}{4\gamma_t^2} \right) \frac{L_{TP}}{\cos(\Phi_s)},
\end{aligned}
\tag{6}
$$

expenses for fabrication, $C_2$, depending on the entire volume of welds, $c_2$:

$$
\begin{aligned}
c_2(\boldsymbol{x}) =& 2N_L \pi D_L \sum_{i=1}^{N_X} \left( \beta_i \left( \frac{D_L^2 \tau_i^2}{8\gamma_i^2} + \frac{t_0 D_L \tau_i}{2\sqrt{2}\gamma_i} \right) \left( \sqrt{\frac{1}{2\sin^2(\psi_{1,i})} + \frac{1}{2}} + \sqrt{\frac{1}{2\sin^2(\psi_{2,i})} + \frac{1}{2}} + \sqrt{\frac{1}{2\sin^2(\psi_{3,i})} + \frac{1}{2}} \right) \right) \\
&+ 2x_{MB} N_L \pi D_L \beta_b \left( \frac{D_L^2 \tau_b^2}{8\gamma_b^2} + \frac{t_0 D_L \tau_b}{2\sqrt{2}\gamma_b} \right) \\
&+ N_L \pi D_L \sum_{i=1}^{N_X} \left( \frac{D_L^2 \min\left(\frac{1}{\gamma_i^2}, \frac{1}{\gamma_{i+1}^2}\right)}{8} + \frac{D_L t_0 \min\left(\frac{1}{\gamma_i}, \frac{1}{\gamma_{i+1}}\right)}{2\sqrt{2}} \right),
\end{aligned}
\tag{7}
$$

5 coating costs, $C_3$, depending on the outer surface area of all tubes, $c_3$:

$$
\begin{aligned}
c_3(\boldsymbol{x}) =& 2N_L \pi D_L \sum_{i=1}^{N_X} \left( \beta_i \sqrt{\frac{L_i^2}{\cos^2(\Phi_p)} + (R_i + R_{i+1})^2 \sin^2\left(\frac{\vartheta}{2}\right)} \right) + x_{MB} N_L \pi D_L \beta_b \left( 2R_1 \sin\left(\frac{\vartheta}{2}\right) \right) \\
&+ N_L \pi D_L \frac{L}{\cos(\Phi_s)},
\end{aligned}
\tag{8}
$$

costs for the transition piece, $C_4$, proportional to the product of head radius and number of jacket legs, $c_4$:

$$
c_4(\boldsymbol{x}) = N_L R_{foot} \xi,
\tag{9}
$$

expenses for transport, $C_5$, expressed by the mass-dependent factor, $c_5$:

10 $$c_5(\boldsymbol{x}) = c_1(\boldsymbol{x}),
\tag{10}$$

and installation costs, $C_6$, modeled by a factor only depending on the number of jacket legs, $c_6$:

$$
c_6(\boldsymbol{x}) = N_L,
\tag{11}
$$

Fixed expenses, $C_7$, are not dependent on any jacket parameter at all. Therefore, the factor, $c_7$, simply takes

$$
c_7(\boldsymbol{x}) = 1.
\tag{12}
$$

In these equations, $\vartheta$ is the angle enclosed by two jacket legs:

$$\vartheta = \frac{2\pi}{N_L}. \tag{13}$$

Bay heights, $L_i$, intermediate bay heights, $L_{m,i}$, radii, $R_i$, and intermediate radii, $R_{m,i}$, are calculated by the following equations:

$$L_i = \frac{L - L_{OSG} - L_{TP}}{\sum_{n=1}^{N_X} q^{n-i}}, \tag{14}$$

$$L_{m,i} = \frac{L_i R_i}{R_i + R_{i+1}}, \tag{15}$$

$$R_i = R_{foot} - \tan\left(\Phi_s\right)\left(L_{OSG} + \sum_{n=1}^{i-1} L_n\right), \tag{16}$$

$$R_{m,i} = R_{foot} - \tan\left(\Phi_s\right)\left(L_{OSG} + \sum_{n=1}^{i-1} L_n + L_{m,i}\right), \tag{17}$$

with the spatial batter angle, $\Phi_s$:

$$\Phi_s = \arctan\left(\frac{R_{foot}\left(1 - \xi\right)}{L}\right). \tag{18}$$

The interconnecting tube angles, $\psi_{1,i}$, $\psi_{2,i}$, and $\psi_{3,i}$, are

$$\psi_{1,i} = \frac{\pi}{2} - \arctan\left(\frac{R_{foot}\left(1 - \xi\right)\sin\left(\frac{\vartheta}{2}\right)\cos\left(\Phi_p\right)}{L}\right) - \arctan\left(\frac{L_{m,i}}{R_i \sin\left(\frac{\vartheta}{2}\right)\cos\left(\Phi_p\right)}\right), \tag{19}$$

$$\psi_{2,i} = \frac{\pi}{2} + \arctan\left(\frac{R_{foot}\left(1 - \xi\right)\sin\left(\frac{\vartheta}{2}\right)\cos\left(\Phi_p\right)}{L}\right) - \arctan\left(\frac{L_{m,i}}{R_i \sin\left(\frac{\vartheta}{2}\right)\cos\left(\Phi_p\right)}\right), \tag{20}$$

$$\psi_{3,i} = 2\arctan\left(\frac{L_{m,i}}{R_i \sin\left(\frac{\vartheta}{2}\right)\cos\left(\Phi_p\right)}\right), \tag{21}$$

with the planar batter angle, $\Phi_p$:

$$\Phi_p = \arctan\left(\frac{R_{foot}\left(1 - \xi\right)\sin\left(\frac{\vartheta}{2}\right)}{L}\right). \tag{22}$$

$\gamma_i$, $\beta_i$, and $\tau_i$ represent the ratios of leg radius-to-thickness, brace-to-leg diameter, and brace-to-leg thickness of the $i$th bay, respectively, obtained by linear stepwise interpolation and counted upwards.

The cost modeling is based on several simplifications and assumptions. The mass-proportional modeling of material costs, $C_1$, is straightforward. Fabrication costs, $C_2$, mainly arise from welding and grinding processes. Although the actual manufacturing processes are quite complex, the entire volume of welds can be considered as a measure of the actual costs. Coating costs, $C_3$ are quite easy to determine by the outer surface area of all tubes, i.e., the area to be coated. There may be synergy effects when coating larger areas, but these are neglected. The expenses for the (stellar-type) transition piece, $C_4$, are assumed to be proportional to the head radius and the number of legs. There are more detailed approaches for this purpose, but no design

of the transition piece is performed, which requires a simple approach. The determination of transport costs, $C_5$, is very diffi-cult. In this work, a mass-dependent approach was selected, which is, however, a large simplification. The mass-dependence reflects that barges have a limited transport capacity, which is at least to some extent mass-dependent or dependent on factors partially related to mass (like the space on the deck of the barge covered by the jacket). Installation costs, $C_6$, cover both the material and the manufacturing of the foundation and the installation at the wind farm location. In case of a pile foundation, these costs are mainly governed by the number of piles, which is equal to the number of legs. The fixed expenses, $C_7$, are not vital for the solution of the optimization problem, but required to shift the costs to more realistic values by covering expenses for cranes, scaffolds, and so forth.

### 3.3 Structural code checks (constraints)

To check jacket designs – i.e., sets of design variables – for validity concerning fatigue and extreme load resistance, structural design code checks are performed. The standards DNV GL RP-C203 (DNV GL AS, 2016) for fatigue and NORSOK N-004 (NORSOK, 2004) for ultimate limit state checks, respectively, are adopted. Both are widely accepted for practical applications and were used to design the UpWind (Vemula et al., 2010) and INNWIND.EU (von Borstel, 2013) reference jackets.

Commonly, the numerical demand of structural code checks is one of the main problems in jacket optimization. To cover the characteristics of environmental impacts on wind turbines, representative loads are to be used for the load assessment. This involves numerous load simulations to consider all load combinations that might occur, particularly in the fatigue case, where the excitation is extrapolated for the entire turbine lifetime. As not only the number of load simulations but also the duration (in case of time domain simulations) correlates to a high demand on numerical capacity, most approaches deploy very simple load assumptions like one design load case per iteration, as already discussed. Altogether, a high numerical effort is required. Utilizing simplified load assumptions like equivalent static loads, where the substructure decoupled from the overlying structure and all interactions are neglected, depicts, however, a massive simplification in case of a wide range of design variables. On the contrary, a pure simulation-based optimization is not applicable due to the aforementioned reasons.

To face this issue, a surrogate modeling approach based on Gaussian process regression (GPR) is deployed. It was shown previously (Häfele et al., 2018a) that good regression results can be obtained by GPR for this purpose. In addition, the regression process relies on a mathematical process that can be interpreted easily and adapted to prior knowledge of the underlying physics. In the present case, the procedure is as follows: A load set with a defined number of design load cases is the basis for structural code checks. The size of the load sets and parameters of environmental and operational conditions are predetermined so as to represent the loads on the turbine adequately. With these load sets, numerical simulations are performed with the aero-hydro-servo-elastic simulation code FAST[6] to obtain output data for the input space of the surrogate model. As this procedure requires much computational effort, the input space is limited to 200 jacket samples[7] (excluding validation samples) in each case as a basis for both surrogate models (fatigue and ultimate limit state), obtained by a Latin hypercube sampling of the input space. In both cases, the results are vectors of output variables, where each element corresponds to a row in the matrix

---

[6]FASTv8 (National Wind Technology Center Information Portal, 2016) was used for this study.

[7]All parameters of these jacket samples are given in the publication, where the surrogate modeling approach was reported (Häfele et al., 2018a).

of inputs, comprising parameters of the input space. Both (input matrix and output vector) the training data. For each new sample, the corresponding output (result of a structural code check) is evaluated by GPR[8]. The specific surrogate models for the considered test problems were derived in a previous work (Häfele et al., 2018a), which revealed that a Matérn 5/2 kernel function is well-suited for the present application.

### 3.3.1 Fatigue limit state

The evaluation of fatigue limit state code checks requires many simulations considering DLC 1.2 and 6.4 production load cases according to IEC 61400-3 (International Electrotechnical Commision, 2009). Under defined conditions (5 MW turbine, 50 m water depth, FINO3 environmental conditions), the required number of design load cases with respect to uncertainty was analyzed in previous papers (Häfele et al., 2018b, 2017). In these works, a load set with 2048 design load cases was gradually reduced to smaller load sets. A reduced load set with 128 design load cases turned out to be a good compromise between accuracy, as the uncertainty arising from the load set reduction is acceptable in this case, and numerical effort, which is significantly smaller compared to the initial load set. I.e., considering two X-joint positions, the standard deviation of fatigue damages increases by a factor of approximately 4 in case of a sixteen-fold load set reduction (from 2048 to 128 design load cases). The actual fatigue assessment involves time domain simulations, an application of stress concentration factors according to Efthymiou (1988)[9] to consider the amplification of stresses due to the geometry of tubular joints, a rainflow cycle counting, and a lifetime prediction by S-N curves and linear damage accumulation. The output value $h_{FLS}$ is the most critical fatigue damage among all damage values of the entire jacket (evaluated in eight circumferential points around each weld), normalized by the calculated damage at design lifetime. A design lifetime of 30 years is assumed, from which 25 years are the actual lifetime of the turbine and 5 years are added to consider malicious fatigue damages during the transport and installation process. Moreover, a partial safety factor of 1.25 is considered in the fatigue assessment.

### 3.3.2 Ultimate limit state

The standard IEC 61400-3 (International Electrotechnical Commision, 2009) requires several design load cases to perform structural code checks for the ultimate limit state. However, not every design load case is critical for the design of a jacket substructure. The relevant ones were analyzed and found to be DLC 1.3 (extreme turbulence during production), 1.6 (extreme sea state during production), 2.3 (grid loss fault during production), 6.1 (extreme sea state during idle), and 6.2 (extreme yaw error during idle) for a turbine with a rated power of 5 MW, under FINO3 environmental conditions and water depth of 50 m. Extreme load parameters are derived by the block maximum method (see Agarwal and Manuel, 2010), where the environmental data is divided into many segments featuring similar distributed data. From this data set, the maximum values are extracted. Based on these maxima, return values (as required by IEC 61400-3) of environmental states are computed. To conduct the structural code checks for the ultimate limit state, time domain simulations are performed and evaluated with respect to the

---

[8]For the background theory of GPR, the reader is referred to Rasmussen and Williams (2008), being the standard reference in this field.

[9]It has to be stated that there are several ways to determine stress concentration factors for tubular joints. This is the approach proposed by the standard DNV GL RP-C203 (DNV GL AS, 2016).

extreme load of the member, where the highest utilization ratio occurs. The result $h_{ULS}$ is a value that approaches 0 in case of infinite extreme load resistance and 1 in case of equal resistance and loads, implying that values greater than 1 are related to designs not fulfilling the ultimate limit state code check. The procedure considers combined loads with axial tension, axial compression, and bending, with and without hydrostatic pressure, which may lead to failure modes like material yielding, overall column buckling, local buckling, or any combination of these. A global buckling check is not performed in this study, as it is known to be uncritical for jacket substructures (Oest et al., 2016).

## 4 Optimization approach and solution methods

The optimization problem incorporates a mixed-integer formulation (due to discrete numbers of legs and bays of the jacket). In order to address this issue, the mixed-integer problem is transferred to multiple continuous problems by solving solutions with a fixed number of legs and bays. As only a few combinations of these discrete variables are considered as realistic solutions for practical applications, this procedure leads to a very limited number of subproblems, but eases the mathematical optimization process significantly. Furthermore, the optimization problem is generally nonconvex, i.e., a local minimum in the feasible region satisfying the constraints is not necessarily a global solution. This is addressed by repeating the optimization with multiple starting points.

The development of new or improved optimization methods solve the numerical optimization problem is not in the scope of this work, because there are methods presented in literature that are known to be suitable for this purpose. Metaheuristic algorithms like Genetic Algorithms or Particle Swarm Optimization are not considered in this work, because they are known to be slow. With regard to efficiency and accuracy, two methods are considered as most powerful for optimization involving nonlinear constraints: sequential quadratic programming (SQP) and interior-point (IP) methods (Nocedal and Wright, 2006). SQP methods are known to be efficient, when the numbers of constraints and design variables are in the same order of magnitude. An advantage is that these methods converge usually better, when the problem is badly scaled. In theory, IP methods have better convergence properties and often outperform SQP methods on large-scale or sparse problems. In this work, both approaches are used to solve the jacket optimization problem[10]. They are outlined briefly in the following.

### 4.1 Sequential quadratic programming method

In principle, SQP can be seen as an adaption of Newton's method to nonlinear constrained optimization problems, computing the solution of the Karush-Kuhn-Tucker equations (necessary conditions for constrained problems). Here, a common approach is deployed, based on the works of Biggs (1975), Han (1977), and Powell (1978a, b). In the first step, the Hessian of the so-called Lagrangian (a term incorporating the objective and the sum of all constraints weighted by Lagrange multipliers) is approximated by the BFGS method (Fletcher, 1987). In the next step, a quadratic programming subproblem is built, where the Lagrangian is approximated by a quadratic term and linearized constraints. This subproblem can be solved by any method

---

[10]The function *fmincon* in MATLAB R2017b was used for this study.

being able to solve quadratic programs. An active-set method described by Gill et al. (1981) is deployed for this task. The procedure is repeated until convergence is reached.

## 4.2 Interior-point method

IP methods are barrier methods, i.e., the objective is approximated by a term that incorporates a barrier term, expressed by a sum of logarithmized slack variables. The actual problem itself, just like in SQP, is solved as a sequence of subproblems. In this work, an approach is deployed, which may switch between line search and trust region methods to approximated problem, depending of the success of each step. If the line search step fails, i.e., when the projected Hessian is not positive definite, the algorithm performs a trusted region step, where the method of conjugate gradients is deployed. The algorithm is described in detail by Waltz et al. (2006).

## 5 Jacket comparison study

In this section, the proposed approach is applied to find and compare optimal jacket designs for the NREL $5\,\mathrm{MW}$ reference turbine (Jonkman et al., 2009). The environmental conditions are adopted from measurements recorded at the research platform FINO3 in the German North Sea.

### 5.1 Reference turbine

The NREL $5\,\mathrm{MW}$ reference turbine, which was published almost one decade ago as a proposal to establish a standardized turbine for scientific purposes, is still an object of many studies in literature dealing with intermediate to high power offshore wind applications. In fact, the market already provides turbines with $8\,\mathrm{MW}$ and aims for even higher ratings. Choosing this reference turbine is motivated by its excellent documentation and accessibility.

The rotor has a hub height of $90\,\mathrm{m}$ and the rated wind speed is $11.4\,\mathrm{m\,s^{-1}}$, where the rotor speed is $12.1\,\mathrm{min^{-1}}$. This is equal to 1P- and 3P-excitations of $0.2\,\mathrm{Hz}$ and $0.6\,\mathrm{Hz}$, respectively. The critical first fore-aft and side-side bending eigenfrequencies of the entire structure are about $0.35\,\mathrm{Hz}$ and do not differ very much when considering only reasonable structural designs for the jacket, because the modal behavior is strongly driven by the relatively soft tubular tower.

### 5.2 Environmental conditions and design load sets

Due to excellent availability, the environmental data is derived from measurements taken from the offshore research platform FINO3, located in the German North Sea close to the wind farm alpha ventus. Compared to the environmental conditions documented in the UpWind design basis (Fischer et al., 2010), the FINO3 measurements are much more comprehensive and allow for a better estimation of probability density functions as inputs for the determination of probabilistic loads (Hübler et al., 2017). The probabilistic load set, which is based on probability density functions of environmental state parameters and reduced in size compared to full load sets used by industrial wind turbine designers, was described in recent studies (Häfele et al., 2018b, 2017). However, there are two drawbacks that have to be mentioned when using this data. First, the FINO3 platform was built

at a location with a quite shallow water depth of $22\,\mathrm{m}$, though the jacket is supposed to be an adequate substructure for water depths above $40\,\mathrm{m}$ and the design water depth in this study is $50\,\mathrm{m}$. Nevertheless, this procedure was also performed in the UpWind project for the design of the OC4 jacket, where the K13 deep water site was considered. Second, the soil properties of the Offshore Code Comparison Collaboration (OC3) (Jonkman and Musial, 2010) are adopted to compute foundation inertias and stiffnesses, as these values are unknown for the FINO3 location. Moreover, it is assumed that the structural behavior of the OC4 jacket pile foundation is valid for all jacket designs, even with varying leg diameters and thicknesses.

## 5.3 Boundaries of design variables and other parameters

The boundaries are chosen conservatively by means of quite narrow design variable ranges (see Table 1), i.e., meaningful parameters that do not exhaust the possible range given by the structural code checks, in a realistic range around the values of the OC4 jacket (Popko et al., 2014). Only three- or four-legged structures with three, four, and five bays are considered as valid solutions for this study. The fixed design variables are, if possible, adopted from the OC4 jacket, which can be seen as a kind of reference structure in this case. The material is steel (S355) with Young's modulus of $210\,\mathrm{GPa}$, shear modulus of $81\,\mathrm{GPa}$, and a density of $7850\,\mathrm{kg\,m^{-3}}$. According to DNV GL AS (2016), an S-N curve with an endurance stress limit of $52.63 \times 10^6\,\mathrm{N\,m^{-2}}$ at $10^7$ cycles and slopes of 3 and 5 before and after endurance limit (curve T), respectively, is applied. The cost model parameters or unit costs, respectively, are adopted from the mean values given in (Häfele et al., 2018a) and set to $a_1 = 1.0\,\mathrm{kg^{-1}}$ (material), $a_2 = 4.0 \times 10^6\,\mathrm{m^{-3}}$ (fabrication), $a_3 = 1.0 \times 10^2\,\mathrm{m^{-2}}$ (coating), $a_4 = 2.0 \times 10^4\,\mathrm{m^{-1}}$ (transition piece), $a_5 = 2.0\,\mathrm{kg^{-1}}$ (transport), $a_6 = 2.0 \times 10^5$ (installation), and $a_7 = 1.0 \times 10^5$ (fixed). With these values, the cost function returns a dimensionless value, also interpretable as capital expenses in €.

## 5.4 Results and discussion

To resolve the mixed-integer formulation of the optimization problem into continuous problems, six subproblems with three legs and three bays ($N_L = 3$, $N_X = 3$), three legs and four bays ($N_L = 3$, $N_X = 4$), three legs and five bays ($N_L = 3$, $N_X = 5$), four legs and three bays ($N_L = 4$, $N_X = 3$), four legs and four bays ($N_L = 4$, $N_X = 4$), and four legs and five bays ($N_L = 4$, $N_X = 5$) were solved using the SQP and IP methods. Therefore, multiple solutions are discussed and compared in the following. The optimization problem is nonconvex, i.e., a local minimum in the feasible region satisfying the constraints is not necessarily a global solution. In theory, both algorithms converge from remote starting points. However, to guarantee global convergence to some extent, all six combinations of fixed integer variables were solved using 100 randomly chosen starting points. Installation costs and fixed expenses were excluded from the objective function and included again after the optimization procedure, because these terms do not have an effect on the individual optimization problems[11]. Gradients were computed by finite differences. The optimization terminated, when the first-order optimality and feasibility measures were both less than $1 \times 10^{-6}$. There was no limit of maximum number of iterations.

The optimal solutions of all six subproblems do not depend on the starting point when using both optimization methods, because there is only one array of optimal design variables in each case. The convergence behavior of both optimization

---

[11]The values shown in the following include all cost terms. The exlusion is only performed during optimization.

**Table 1.** Boundaries of jacket model parameters for design of experiments. Topological, tube sizing, and material parameters are separated in groups, single values state that the corresponding value is held constant.

| Parameter | Description | Lower Boundary | Upper Boundary |
|---|---|---|---|
| $N_L$ | Number of legs | 3 | 4 |
| $N_X$ | Number of bays | 3 | 5 |
| $R_{foot}$ | Foot radius | $6.792\,\mathrm{m}$ | $12.735\,\mathrm{m}$ |
| $\xi$ | Head-to-foot radius ratio | 0.533 | 0.733 |
| $L$ | Entire jacket length | $70.0\,\mathrm{m}$ | |
| $L_{MSL}$ | TP elevation over MSL | $20.0\,\mathrm{m}$ | |
| $L_{OSG}$ | Lowest leg segment height | $5.0\,\mathrm{m}$ | |
| $L_{TP}$ | TP segment height | $4.0\,\mathrm{m}$ | |
| $q$ | Ratio of two consecutive bay heights | 0.640 | 1.200 |
| $x_{MB}$ | Mud brace flag | true (1) | |
| $D_L$ | Leg diameter | $0.960\,\mathrm{m}$ | $1.440\,\mathrm{m}$ |
| $\gamma_b$ | Leg radius-to-thickness ratio (bottom) | 12.0 | 18.0 |
| $\gamma_t$ | Leg radius-to-thickness ratio (top) | 12.0 | 18.0 |
| $\beta_b$ | Brace-to-leg diameter ratio (bottom) | 0.533 | 0.800 |
| $\beta_t$ | Brace-to-leg diameter ratio (top) | 0.533 | 0.800 |
| $\tau_b$ | Brace-to-leg thickness ratio (bottom) | 0.350 | 0.650 |
| $\tau_t$ | Brace-to-leg thickness ratio (top) | 0.350 | 0.650 |
| $E$ | Material Young's modulus | $2.100 \times 10^{11}\,\mathrm{N\,m^{-2}}$ | |
| $G$ | Material shear modulus | $8.077 \times 10^{10}\,\mathrm{N\,m^{-2}}$ | |
| $\rho$ | Material density | $7.850 \times 10^{3}\,\mathrm{kg\,m^{-3}}$ | |

methods is illustrated in Figure 2, where the OC4 jacket with varying numbers of legs and bays was assumed as the starting point. This structure has a foot radius, $R_{foot}$, of $8.79\,\mathrm{m}$, a head-to-foot radius ratio, $\xi$, of $0.67$, and a ratio of two consecutive bay heights, $q$, of $0.8$. Moreover, it has a leg diameter, $D_L$, of $1.2\,\mathrm{m}$, and entirely constant tube dimensions from bottom to top, i.e., leg radius-to-thickness ratios, $\gamma_b$ and $\gamma_t$, of 15, brace-to-leg diameter ratios, $\beta_b$ and $\beta_t$, of $0.5$, and brace-to-leg diameter ratios, $\tau_b$ and $\tau_t$, of $0.5$. The optimization process needed between 30 and 40 iterations using the SQP method and between 50 and 70 iterations using the IP method to converge. It is worth mentioning that the maximum constraint violation (feasibility) of the three-legged designs was higher at the beginning of the optimization process, but converges stably. For the same reason, the four-legged designs have a higher improvement potential compared to the initial solution. The accuracy obtained by both methods is similar. The solutions are all feasible, because they fulfill the Karush-Kuhn-Tucker conditions, and all constraint violations are around zero. Therefore, the optima are probably global optima for the given design variable boundaries.

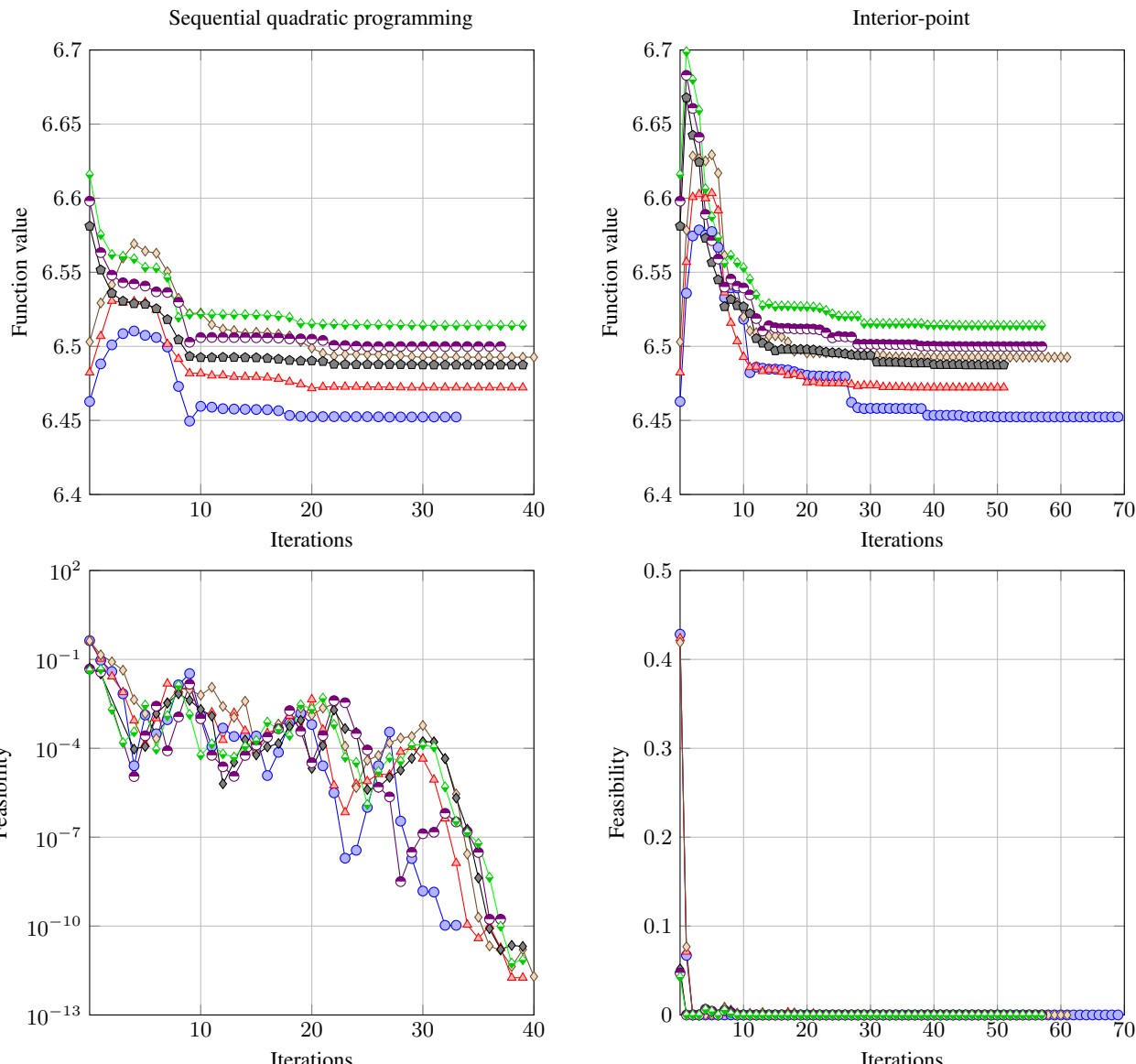

**Figure 2.** Function and feasibility (maximum constraint violation) values during optimization procedure of all six subproblems ($-\!\circ\!-$: jacket with three legs and three bays, $-\!\triangle\!-$: jacket with three legs and four bays, $-\!\diamond\!-$: jacket with three legs and five bays, $-\!\bullet\!-$: jacket with four legs and three bays, $-\!\circ\!-$: jacket with four legs and four bays, $-\!\diamond\!-$: jacket with four legs and five bays). Starting point (iteration "0") is the OC4 jacket with varying number of legs and bays in all cases. One iteration involves 11 evaluations of the objective function and the nonlinear constraints.

The optimal solutions obtained by the sequential quadratic programming method[12] are illustrated in Table 2. Additionally, the topologies of all optimal solutions are shown in Figure 3. With respect to the constraints and presumptions of this study ($5\,\mathrm{MW}$ turbine, $50\,\mathrm{m}$ water depth, given environmental conditions and cost parameters), jackets with three legs are beneficial in terms of capital expenses. The three-legged jacket with three bays ($N_L = 3$, $N_X = 3$) is the best solution, i.e., is related to the lowest total expenditures, among the considered jackets. The solutions show some interesting specialties. The foot radii, $R_{foot}$, are at the upper boundaries in case of the three-legged structures, while the head-to-foot radius ratios, $\xi$, are at the lower boundaries. Probably, this arises from the combination of cost function and nonlinear constraints, where a large foot radius is quite beneficial, because it generally provides a higher load capacity, while a small head radius is favorable due to lower TP costs. In the four-legged case, the foot radii are lower, but still relatively high. In any case, it seems to be beneficial, when the ratio of two consecutive bay heights, $q$, is slightly below 1 (lower bays are higher than upper bays). Concerning tube dimensions, the leg diameters, $D_L$, are relatively small, in case of the four-legged jackets even at the lower boundary. The structural load capacity is established by high brace diameters (represented by design variables $\beta_b$ and $\beta_t$, values at the bottom and top of the structures both at upper boundaries). The brace thicknesses, represented by $\tau_b$ and $\tau_t$, show intermediate values in the range of design variables, while the values for $\tau_t$ are higher in case of three-legged designs. Moreover, the structural resistance is strongly driven by the leg thicknesses. While the optimal values of $\gamma_b$ are low in each case, implying high leg thicknesses at the jacket bottom, the values of $\gamma_t$ are much higher. The impact of all design variables on the objective function is easier to understand, when the sensitivities of cost model terms to variations in design variables are considered. In Figure 4, each subplot shows the variation of the total costs, $C_{total}$, and the cost function terms $C_1$ (proportional to $C_5$), $C_2$, $C_3$, and $C_4$ due to a $1\,\%$ one-at-a-time variation of each continuous design variable in three different phases of the optimization process (initial, intermediate, and final phase). The terms $C_6$ and $C_7$ are not impacted by any continuous design variable and therefore not considered. For instance, a $1\,\%$ increase of the foot radius, $R_{foot}$, causes increasing material costs of $\Delta C_1 = 0.14\,\%$, evaluated for the initial design, but increasing material costs of $\Delta C_1 = 0.26\,\%$, evaluated for the optimal design. Therefore, the sensitivity of this cost term varies during the optimization process. In contrast, the variation of transition piece expenses does not change (which is reasonable, because this term only depends linearly on the number of legs, $N_L$, the foot radius, $R_{foot}$, and the head-to-foot radius ratio, $\xi$). In general, Figure 4 shows that there is no design variable with strongly varying impact on any term of the cost function. It can also be concluded that tube sizing variables impact the costs much stronger than topological variables, disregarding the number of legs and bays. Among the considered design variables, the leg diameter, $D_L$, and leg radius-to-thickness ratios, $\gamma_b$ and $\gamma_t$, are design driving (together with the number of legs, $N_L$) due to a significant impact both on the costs and on the structural code checks. In addition, an interesting specialty is featured by the cost term $C_4$, which is only impacted by topological design variables, more precisely the foot radius, $R_{foot}$, and the head-to-foot radius ratio, $\xi$. As a large foot radius, $R_{foot}$, is needed to establish structural resistance, this cost term penalizes large head-to-foot radius ratios, $\xi$. For this reason, this value is at the lower boundary for all design solutions.

Regarding the costs of the jackets, the best solution with three legs and three bays is related to capital expenses of $10^{6.452} = 2\,831\,000$. Altogether, this is a meaningful value and the designs are not far off from structural designs that are known from

---

[12]As the accuracy of the SQP and IP methods are similar here, only results obtained by the SQP method are shown in the following.

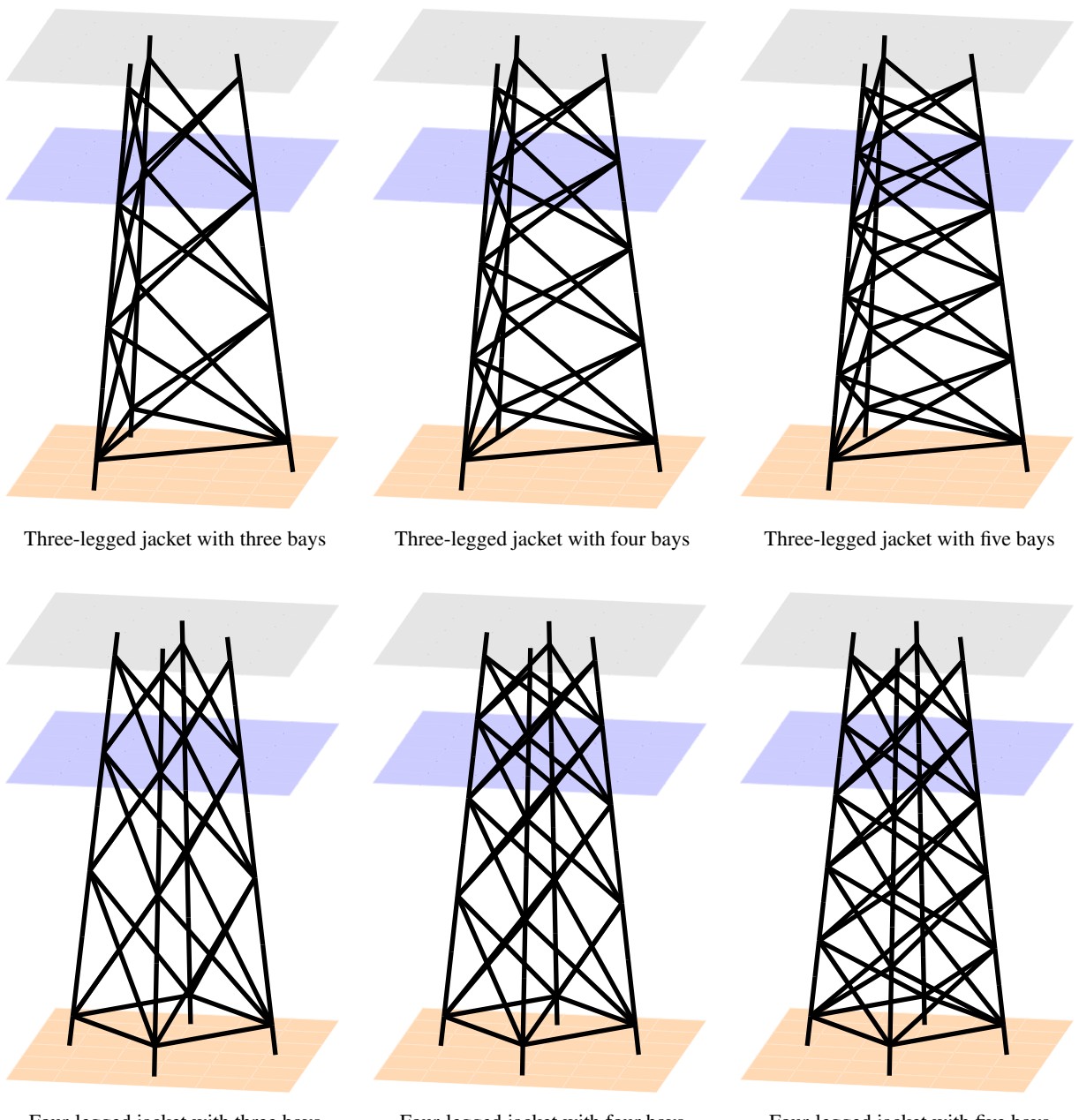

Three-legged jacket with three bays     Three-legged jacket with four bays     Three-legged jacket with five bays

Four-legged jacket with three bays     Four-legged jacket with four bays     Four-legged jacket with five bays

**Figure 3.** Topologies of optimal solutions $x^*$. All images are displayed in the same scale. Line widths are not correlated to tube dimensions.

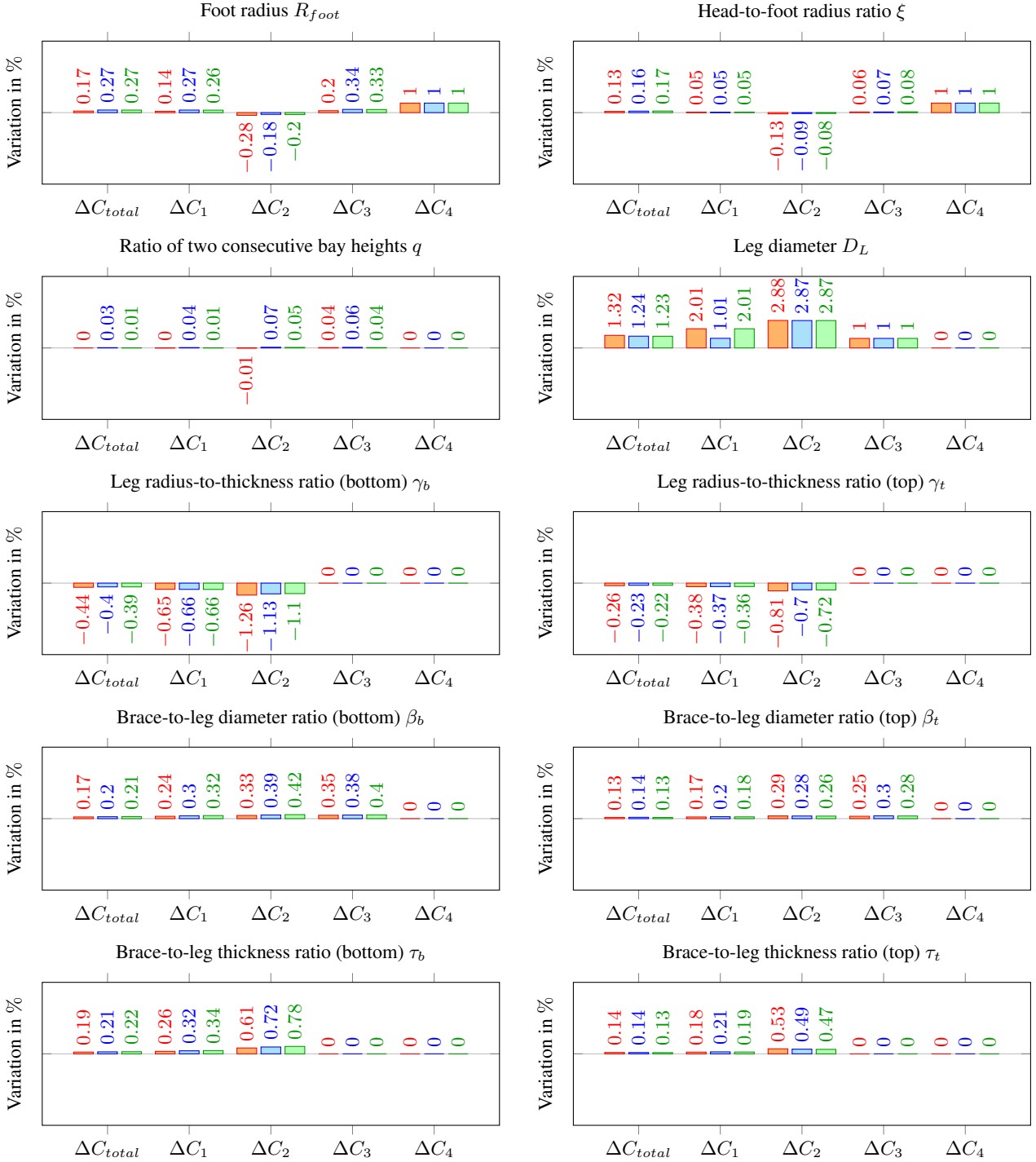

**Figure 4.** Variations of total costs, $\Delta C_{total}$, and cost function terms $\Delta C_1$ (material), $\Delta C_2$ (manufacturing), $\Delta C_3$ (coating), and $\Delta C_4$ (transition piece) due to $1\%$ one-at-a-time variations of design variables (subplots) in $\%$. Derivatives were computed for the initial design ( ), an intermediate design after 15 iterations ( ), and the optimal design ( ) of the three-legged structure with three bays ($N_L = 3$, $N_X = 3$).

**Table 2.** Optimal solutions of design variables $x^*$ obtained by sequential quadratic programming method for fixed values of $N_L$ and $N_X$.

| | | Optimal solution | | | | | |
|---|---|---|---|---|---|---|---|
| | $N_L$ | 3 | 3 | 3 | 4 | 4 | 4 |
| | $N_X$ | 3 | 4 | 5 | 3 | 4 | 5 |
| | $R_{foot}$ in m | 12.735 | 12.735 | 12.735 | 10.894 | 10.459 | 10.549 |
| | $\xi$ | 0.533 | 0.533 | 0.533 | 0.533 | 0.533 | 0.533 |
| | $q$ | 0.937 | 0.941 | 0.936 | 0.813 | 0.809 | 0.977 |
| $x^*$ | $D_L$ in m | 1.021 | 1.021 | 1.023 | 0.960 | 0.960 | 0.960 |
| | $\beta_b$ | 0.800 | 0.800 | 0.800 | 0.800 | 0.799 | 0.787 |
| | $\beta_t$ | 0.800 | 0.800 | 0.800 | 0.800 | 0.800 | 0.800 |
| | $\gamma_b$ | 12.000 | 12.000 | 12.000 | 12.680 | 12.259 | 12.000 |
| | $\gamma_t$ | 16.165 | 16.029 | 15.928 | 18.000 | 18.000 | 18.000 |
| | $\tau_b$ | 0.513 | 0.505 | 0.493 | 0.497 | 0.493 | 0.478 |
| | $\tau_t$ | 0.472 | 0.466 | 0.454 | 0.383 | 0.387 | 0.383 |
| Overall mass in t | | 423 | 444 | 467 | 412 | 426 | 439 |
| $f(x^*) = \log_{10}\left(C_{total}(x^*)\right)$ | | 6.452 | 6.472 | 6.493 | 6.487 | 6.500 | 6.514 |
| $h_1(x^*) = h_{FLS}(x^*) - 1$ | | $1.172 \times 10^{-10}$ | $3.966 \times 10^{-11}$ | $1.151 \times 10^{-10}$ | $1.450 \times 10^{-10}$ | $-1.056 \times 10^{-10}$ | $-1.721 \times 10^{-10}$ |
| $h_2(x^*) = h_{ULS}(x^*) - 1$ | | $7.819 \times 10^{-10}$ | $2.678 \times 10^{-10}$ | $1.093 \times 10^{-10}$ | $3.978 \times 10^{-10}$ | $3.980 \times 10^{-10}$ | $5.995 \times 10^{-10}$ |

practical applications, because it has already been reported in literature that three-legged designs may be favorable in terms of costs (Chew et al., 2014) and three-legged structures have already been built. However, the other solutions are more expensive, but not completely off. As there is some uncertainty in the unit costs, the other jackets may also be reasonable designs under slightly different boundaries. A more detailed cost breakdown is given in Figure 5, which shows the cost contributions of all six structures and where the actual cost savings come from. The lightest structure is the four-legged jacket with three bays, while the three-legged jacket with five bays is the heaviest one, which shows in expenses for material and transport according to the cost model used for this study. Nevertheless, the mass of all structures is quite similar. Other than expected, the jacket with the lowest expenditures for manufacturing is also the four-legged one with three bays and not the three-legged jacket with three bays, which has the least number of joints. The three-legged structures benefit – from the economical point of view – mainly from lower expenses for coating, transition piece, and, most distinctly, installation costs. In total, these contributions add up to lower costs of the three-legged jackets, except the one with five bays ($10^{6.493} = 3\,112\,000$), which is more expensive than the four-legged one with three bays ($10^{6.487} = 3\,069\,000$). The most expensive jackets are the four-legged ones with four ($10^{6.500} = 3\,162\,000$) and five ($10^{6.514} = 3\,266\,000$) bays, where the latter is about 15% more expensive than the best solution among the six sub-solutions. A reasonable option may also be the jacket with three legs and four bays, which features a total cost value of $10^{6.472} = 2\,965\,000$. In total, there is no jacket that is far too expensive compared to the others. It is indeed imaginable to find an appropriate application for each one.

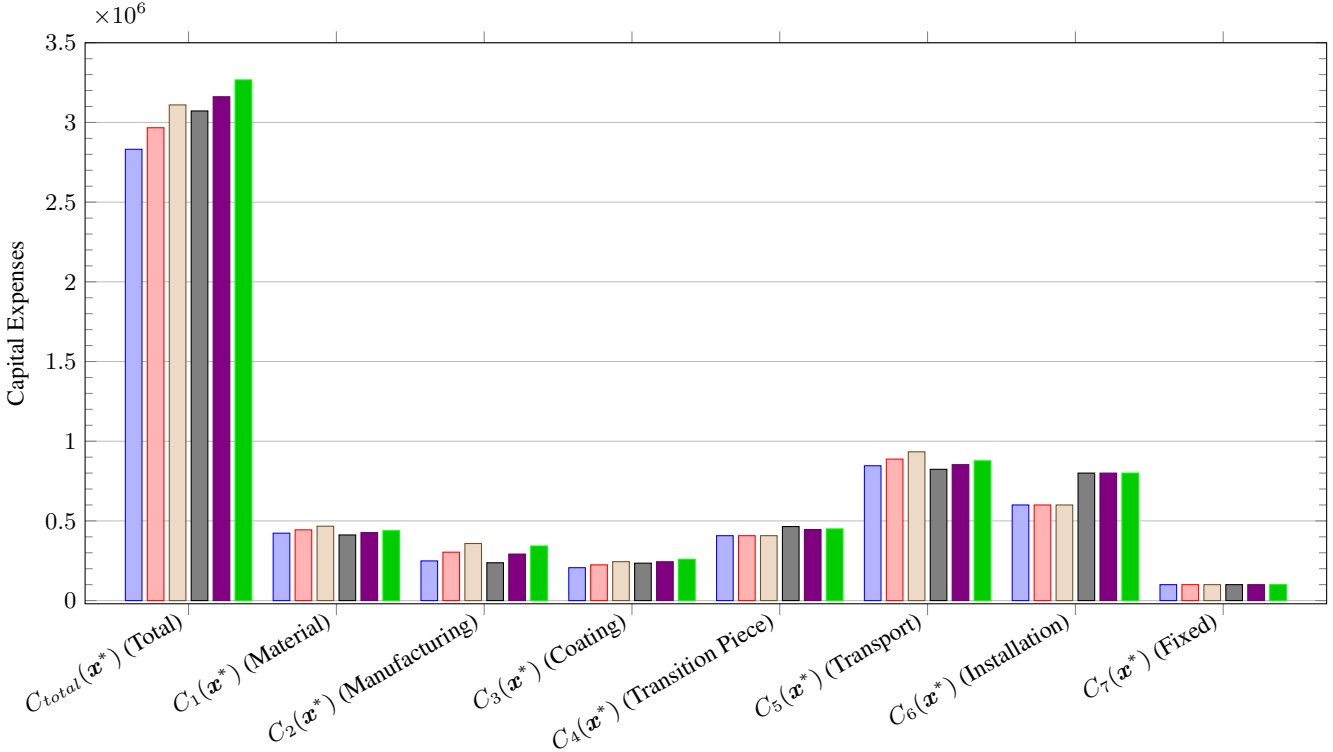

**Figure 5.** Expenses comparison of optimal solutions of three-legged jacket with three bays (▯▮), three-legged jacket with four bays (▯▮), three-legged jacket with five bays (▯▮), four-legged jacket with three bays (▮▮), four-legged jacket with four bays (▮▮), and four-legged jacket with five bays (▮▮).

From the computational point of view, the optimization procedure based on surrogate models is very efficient. The numbers of iterations needed to find an optimal solution (from about 30 to 40 using the SQP method and from about 50 to 70 using the IP method) are related to computation times from about $15\,\mathrm{min}$ to $30\,\mathrm{min}$ on a single core of a work station with Intel Xeon E5-2687W v3 central processing unit and $64\,\mathrm{GB}$ random access memory. Compared to simulation-based approaches, this can be considered as very fast. The number of iterations may be decreased, when using analytical gradients of the objective function, because using finite differences is generally more prone to numerical errors, but is not vital at this level of computational expenses. It has to be pointed out that the training data set of the surrogate models required $200 \times 128 = 25\,600$ time domain simulations in the fatigue and $200 \times 10 = 2000$ in the ultimate limit state case, thus $27\,600$ simulations in total, excluding validation samples. However, for the computation of the training data, a compute cluster was utilized, which allows for the computation of many design load cases in parallel. Therefore, the presented approach based on GPR allows for outsourcing computationally expensive simulations on high-performance clusters, while the closed-loop optimization, which cannot be parallelized completely, can be run on a workstation with lower computational capacity.

**Table 3.** Optimal solutions of design variables $x^*$ obtained by sequential quadratic programming method for fixed values of $N_L$ and $N_X$ using a pure mass-dependent objective function.

| | | Optimal solution | | | | | |
|---|---|---|---|---|---|---|---|
| | $N_L$ | 3 | 3 | 3 | 4 | 4 | 4 |
| | $N_X$ | 3 | 4 | 5 | 3 | 4 | 5 |
| | $R_{foot}$ in m | 12.735 | 12.735 | 12.735 | 12.735 | 12.735 | 12.735 |
| | $\xi$ | 0.533 | 0.533 | 0.533 | 0.533 | 0.533 | 0.533 |
| | $q$ | 1.062 | 0.987 | 0.936 | 1.200 | 1.200 | 1.178 |
| $x^*$ | $D_L$ in m | 1.025 | 1.023 | 1.023 | 0.960 | 0.960 | 0.960 |
| | $\beta_b$ | 0.800 | 0.800 | 0.800 | 0.730 | 0.757 | 0.800 |
| | $\beta_t$ | 0.800 | 0.800 | 0.800 | 0.800 | 0.800 | 0.800 |
| | $\gamma_b$ | 12.000 | 12.000 | 12.000 | 13.194 | 13.318 | 12.000 |
| | $\gamma_t$ | 16.459 | 16.250 | 15.928 | 18.000 | 18.000 | 18.000 |
| | $\tau_b$ | 0.509 | 0.502 | 0.493 | 0.510 | 0.470 | 0.443 |
| | $\tau_t$ | 0.472 | 0.466 | 0.454 | 0.386 | 0.361 | 0.350 |
| Overall mass in t | | 423 | 444 | 467 | 404 | 409 | 454 |
| $f(x^*) = \log_{10}(C_{total}(x^*))$ | | 5.627 | 5.647 | 5.669 | 5.606 | 5.612 | 5.657 |
| $h_1(x^*) = h_{FLS}(x^*) - 1$ | | $-7.149 \times 10^{-12}$ | $6.767 \times 10^{-12}$ | $1.262 \times 10^{-12}$ | $5.047 \times 10^{-13}$ | $2.140 \times 10^{-12}$ | $-1.017 \times 10^{-12}$ |
| $h_2(x^*) = h_{ULS}(x^*) - 1$ | | $4.367 \times 10^{-11}$ | $2.961 \times 10^{-11}$ | $5.087 \times 10^{-12}$ | $2.693 \times 10^{-12}$ | $3.865 \times 10^{-12}$ | $9.948 \times 10^{-13}$ |

The question remains, what happens, when some cost terms were neglected. An associated question is, how the approach performs compared to a pure mass-dependent one, which can be considered as state of the art in jacket optimization. For this purpose, all unit costs except $a_1$ were set to zero and the optimization procedure was repeated using the sequential quadratic programming method. The results, including optimal design variables and resulting values of objective and constraint functions, are shown in Table 3. Under these assumptions, the four-legged jackets are better (in terms of minimal mass) than the three-legged ones. Interestingly, similar design variables are obtained when comparing the values to the ones obtained by the more comprehensive cost model in Table 2, particularly in case of the three-legged jackets. The resulting objective function values are, in relation, similar to the material costs in Figure 5. In other words, a pure mass-dependent cost function approach yields approximately proportional costs, when the installation costs (depending on the number of legs) are considered, and similar designs. The reason for this is that all cost terms $C_1 \dots C_5$ depend in some way on the tube dimensions and the topology does not impact the costs to a great extent, as seen in Figure 4. Indeed, the largest proportion of costs is purely mass-dependent, as the factors $c_1$ and $c_5$ are the mass of the structure. Therefore, the proposed cost model can lead to more accurate results, but a mass-dependent approach would be sufficient to draw the same conclusions.

# 6 Benefits and limitations of the approach

With respect to the state of the art, the present approach can be considered as the first one addressing the jacket optimization problem holistically, which incorporates four main improvements: a detailed geometry model with both topological and tube sizing design variables, an analytical cost model based on the main jacket cost contributions, sophisticated load assumptions and assessments, and the treatment of results in the way that the optimization problem is rather seen as a methodology for early design stages. All these points lead to a better understanding how to address the multidisciplinary design optimization problem and to much more reliable results.

However, some drawbacks and limitations remain, which have to be considered when dealing with the results of this study. In general, the approach is easy to use, also in industrial applications, but needs some effort in implementation. Furthermore, the present study does not incorporate a completely reliability-based design procedure, which is not beyond the means when using Gaussian process regression to perform structural code checks. However, it is still a matter of research, how safety factors can be replaced by a meaningful probabilistic design and quite simple to advance the present approach to a robust one. In order to reduce the numerical cost (in particular concerning the number of time domain simulations needed to sample the input design space for surrogate modeling of structural code checks), the number of design variables is limited. The application of GPR as a machine learning approach to evaluate structural code checks performs numerically fast, but requires indeed numerous time domain simulations to generate training and validation data sets. This is beneficial when dealing with numerically expensive studies (as it is in this case), but might lead to numerical overhead when only considering one jacket design. Care has to be taken when transferring the results to designs with more sophisticated geometry. Moreover, the parameterization of cost and structural code check models is site- and turbine-dependent. Therefore, the outcome of this study might not be directly transferable to other boundaries, but requires recalculations. In particular, the utilized design standards and structural code checks are known to be conservative. The cost model has also shortcomings to be mentioned. Some costs are affected by uncertain or indeterminable impacts. There is a number of examples. Transport and installation costs are strongly dependent on availability of barges or vessels. The uncertainty in weather conditions can affect transport and installation costs. Furthermore, the design may be directly impacted, if production facilities are not available. All these effects are not considered in the cost model.

In addition, it is important to highlight again that this study does not provide a detailed design methodology, but an approach to obtain preliminary decision guidance in the earliest wind farm planning stage. This is actually not a limitation, but has to be considered, when dealing with the results of this study. There are indeed many studies known from literature that address the tube dimensioning problem in larger extension. However, these approaches assume that the structural topology is always optimal, even in case of significant variations in tube dimensions. For instance, all optimal jackets have a larger bottom width than the OC4 jacket, while the design driving leg diameters are relatively small. This indicates that topological design variables with minor impact on costs are useful factors to establish the structural resistance.

# 7  Conclusions

The present work was introduced by four main points to be considered in order to improve the state of the art in the field of jacket optimization. The first one, the treatment of the jacket design problem in terms of a holistic topology and tube sizing problem instead of a pure tube dimensioning problem was addressed by a 20-parameter jacket model, from which twelve parameters are design variables. The second, important point leads to the utilization of a more complex (compared to mass-dependent), nevertheless easy to handle, cost model. In order to face the challenging task of numerically efficient structural code check evaluations, a machine learning approach based on Gaussian process regression was applied as the third point. On this basis, gradient-based optimization was deployed to find optimal design solutions. Last, optimization results were considered differently compared to approaches presented in literature. It was pointed out that the solution is not supposed to be the final design, but a very good starting point to find an initial solution for exact tube dimensioning.

The conclusions of this work are manifold. From the numerical point of view, surrogate modeling seems – as matters stand today – to be the most promising approach enabling to address the computationally very expensive jacket optimization problem efficiently, because other approaches in literature go along with massive simplifications, mainly in load assumptions. The optimization methods that were used to find the optimal solution seem to be appropriate for the given problem, even in terms of finding a global optimum. The present paper does not provide improvements of state-of-the-art gradient-based optimization, but active-set SQP and IP methods both converge efficiently and accurately on the given problem.

From the application-oriented point of view, it can be stated that three-legged jackets with only three bays depict the best solution (in terms of costs) for offshore turbines with about $5\,\mathrm{MW}$ rated power in $50\,\mathrm{m}$ water depth, which confirms the results from other studies in literature. Due to the cost model, the additional load bearing capacity gained by the extra leg of a four-legged structure cannot compensate the higher costs arising from several cost factors directly related to the number of legs. By contrast, it is rather beneficial to increase the tube dimensions and maintain the number of structural elements on a minimum level. It was shown that the same results were obtained, when using a mass-dependent cost function, also considering the number of jacket legs.

Heading to turbines with higher rated power or installations in deeper waters, the proposed methodology might lead to the result that the best jacket solution for this case looks completely different. Before this can be analyzed, simulation tools need to be improved to enable the consideration of nonlinear effects for rotors with very high diameter and innovative control strategies.

*Competing interests.*  No competing interests are present in this study.

*Acknowledgements.*  This work was supported by the compute cluster which is funded by the Leibniz Universität Hannover, the Lower Saxony Ministry of Science and Culture (MWK), and the German Research Foundation (DFG).

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

## Appendix A: Nomenclature

| | |
|---|---|
| DLC | Design load case |
| IP | Interior-point method |
| SQP | Sequential quadratic programming method |
| $\Phi_p$ | Planar (two-dimensional) batter angle |
| $\Phi_s$ | Spatial (three-dimensional) batter angle |
| $\beta_b$ | Brace-to-leg diameter ratio at bottom (jacket model parameter) |
| $\beta_i$ | Brace-to-leg diameter ratio in the $i$th bay |
| $\beta_t$ | Brace-to-leg diameter ratio at top (jacket model parameter) |
| $\gamma_b$ | Leg radius-to-thickness ratio at bottom (jacket model parameter) |
| $\gamma_i$ | Leg radius-to-thickness ratio in the $i$th bay |

| $\gamma_t$ | Leg radius-to-thickness ratio at top (jacket model parameter) |
|---|---|
| $\xi$ | Head-to-foot radius ratio (jacket model parameter) |
| $\rho$ | Material density (jacket model parameter) |
| $\vartheta$ | Angle enclosed by two jacket legs |
| $\tau_b$ | Brace-to-leg thickness ratio at bottom (jacket model parameter) |
| $\tau_i$ | Brace-to-leg thickness ratio in the $i$th bay |
| $\tau_t$ | Brace-to-leg thickness ratio at top (jacket model parameter) |
| $\psi_{1,i}$ | Lower brace-to-leg connection angle in the $i$th bay |
| $\psi_{2,i}$ | Upper brace-to-leg connection angle in the $i$th bay |
| $\psi_{3,i}$ | Brace-to-brace connection angle in the $i$th bay |
| $C_j$ | Expenses related to $j$th cost factor |
| $C_{total}$ | Total capital expenses |
| $D_{Bb}$ | Bottom brace diameter |
| $D_{Bt}$ | Top brace diameter |
| $D_L$ | Leg diameter (jacket model parameter) |
| $E$ | Material Young's modulus (jacket model parameter) |
| $G$ | Material shear modulus (jacket model parameter) |
| $L$ | Overall jacket length (jacket model parameter) |
| $L_{MSL}$ | Transition piece elevation over MSL (jacket model parameter) |
| $L_{OSG}$ | Lowest leg segment height (jacket model parameter) |
| $L_{TP}$ | Transition piece segment height (jacket model parameter) |
| $L_i$ | $i$th jacket bay height |
| $L_{m,i}$ | Distance between the lower layer of K joints and the layer of X joints of the $i$th bay |
| $N_L$ | Number of legs (jacket model parameter) |
| $N_X$ | Number of bays (jacket model parameter) |
| $R_{Foot}$ | Foot radius (jacket model parameter) |
| $R_{Head}$ | Head radius |
| $R_i$ | $i$th jacket bay radius at lower K joint layer |
| $R_{m,i}$ | Radius of the $i$th X joint layer |
| $T_{Bb}$ | Bottom brace thickness |
| $T_{Bt}$ | Top brace thickness |
| $T_{Lb}$ | Bottom leg thickness |
| $T_{Lt}$ | Top leg thickness |
| $a_j$ | $j$th unit cost |
| $c_j$ | $j$th cost factor |

| | |
|---|---|
| $f$ | Objective function value |
| $h_1$ | First inequality constraint value |
| $h_2$ | Second inequality constraint value |
| $h_{FLS}$ | Maximal normalized tubular joint fatigue damage |
| $h_{ULS}$ | Maximal extreme load utilization ratio |
| $q$ | Ratio of two consecutive bay heights (jacket model parameter) |
| $\boldsymbol{x}$ | Array of design variables |
| $\boldsymbol{x}_{lb}$ | Array of lower boundaries |
| $x_{MB}$ | Mud brace flag (jacket model parameter) |
| $\boldsymbol{x}_{ub}$ | Array of upper boundaries |