# Peer review of "A comparison study on jacket substructures for offshore wind turbines based on optimization"

_Wind Energy Science, 2018_

## Referee Comment (RC1) · Anonymous Referee #1 · 17 Oct 2018

**General comments:**

The authors present a fairly comprehensive framework for cost optimization of jacket designs. This framework includes variables controlling topology (both discrete and continuous ones), more involved cost models than simply the total mass and approaches for assessment of both fatigue damage and extreme loads that are more accurate than the limited number of load cases usually studied in support structure optimization. All this while keeping the computational effort at a tractable level. Seen in isolation, this already represents a certain amount of progress compared to state-of-the-art jacket optimization approaches. While there are certain limitations present (for example the number of design variables), these are generally not limitations of the framework, but simply limitations of the application of that framework in the current study. In general,

the presented approach seems well founded and seems to be computationally well behaved. Hence, the framework seems like a potentially promising methodology for future jacket optimization endeavors. However, there are some key features that are missing and need to be addressed. Especially related to the justification of the framework (the cost function and the variables affecting design topology) when compared to previous work, based on the specific results of the study rather than more general ideas about the design process. While some examples will be addressed specifically below, in general the paper could also use a bit of revision language-wise.

**Specific comments:**

Abstract:

- On page 1, lines 12-13, you write: "The approach shows reasonable and promising results, ..." Even for the abstract, this is a little vague. You should say something about how the method seems to succeed in finding global minima with reasonable computational effort and how the method allows for a systematic way of comparing cost and structural performance for optimal designs with different topologies.

1. Introduction:

- On page 2 you have a fairly comprehensive literature review. You might also want to consider adding a note about Sandal et al (2018, Marine Structures) where the effect of varying leg distance (functionally equivalent to what you call leg radius) was studied to a certain extent.

2. Problem statement:

- On page 4, lines 2-4, you write about the limited effect of changes to eigenfrequencies for the design. This would likely be a larger (if not necessarily critical) issue if the soil and foundation were modeled in more detail than what you have done in this study. Though you correctly note that any design close to resonance would fail the fatigue check, there are practical reasons (related to the numerical behavior of the

optimization process) that can make it beneficial to include explicit constraints on the eigenfrequency rather than relying exclusively on the fatigue constraints.

3. Objective and constraints:

- On page 7, line 10, you define $c_7$. This is constant for all designs and hence has no effect on the optimization process. While it may be instructive to note the existence of such additional costs in practice, for the optimization problem being solved such a term is irrelevant. Hence, you should at the very least state whether this term is part of the implemented computations or if it is merely added at the end of the optimization.

- On page 9, lines 3-4, you describe a "space-filling sampling of the input space." As far as I can see, the details of this sampling is not given either here or in the previous paper about the surrogate modeling. It would be instructive to have some more details here. At the very least, specifically what kind of sampling method are you using?

- On page 9, lines 13-14, you write "A number of 128 turned out to be a good compromise between accuracy and numerical effort." The details of this was given in previous work, but for the sake of the reader of the present work it would be instructive to give a quick summary of what level of accuracy 128 load cases represents.

- On page 9, lines 16-17, you write "The output value $h_{FLS}$ is the most critical fatigue damage among all damage values of the entire jacket..." Considering the results you present later, this does not seem to have been a problem here, but note that using such a constraint (the maximum of a discrete set) could lead to discontinuities that affect your gradients (sensitivities) adversely.

- On page 9, line 27, you write "Extreme load parameters are derived by the block maximum method according to Agarwal and Manuel (2010)." This needs a little more detail. Does this include a statistical extrapolation to a 50-year (or n-year) return value?

- On page 9, lines 28-29, you write "... evaluated with respect to the extreme load of the member, where the highest utilization ratio occurs." Here you have the same potential
issue as with the fatigue constraint.

5. Jacket comparison study:

- On page 12, lines 18-21, you summarize the different fixed integer design variable combinations you study. Note that term $c_6$ in your previously defined cost function is constant for each of these combinations and hence has no (computational) effect on the individual optimization problems (merely adds a constant term to the solution).

- On page 12, lines 27-33 [also on page 16, lines 3-6], you describe the behavior of your optimization routines. It would be instructive to plot an example of the convergence. For example cost (and maybe feasibility) vs number of iterations. How many function evaluations is typically involved per iteration? The number of function evaluations (in terms of both objective and constraints) is a more direct measure of computational speed/effort than the number of iterations (and is more easily generalized to different machines). It also says something about how the algorithm is behaving.

- On pages 13-14, you discuss the properties of the optimal designs. What do the initial designs look like (especially in terms of cost and feasibility with respect to code checks)? How are the optimal designs compared to a "typical" initial design, maybe compared to the OC4? Is the most optimal design topology $(N_L, N_X)$ = (3,3) also the one with the most improvement compared to "initial" designs? How is this for the various other topologies? Obviously, the results here are only meant to illustrate the method, so the specifics of the optimal designs are not so important. However, it would give more insight into the effect of your cost model and your chosen design variables if you explain in more detail what kind of optimal designs your methodology tends to produce, with more clear reference to initial designs. What would you say are the design driving variables?

- On page 14, line 5, you write "Altogether, this is meaningful and not far off from structural designs that are known from practical applications." Do the authors mean that the designs are close to that seen in practical applications or that the costs are?

Please elaborate a bit on this statement.

- On page 16, Figure 3, you show a cost breakdown of the optimal designs. Since $C_1$ and $C_5$ both depend on the mass, note that the mass is associated with the largest proportion of the cost, but this is not immediately obvious from the figure. From a practical point of view, these two points indeed represent different aspects of the production and installation process, but since both these terms directly contribute to lighter designs, their effects in an optimization context are the same (and in the figure, these two terms are clearly just scaled versions of each other). While $C_6$ does not impact the optimization directly, its presence in the cost breakdown is justified by how it clearly shows where a significant portion of the difference in total cost between 3- and 4-legged designs come from. The importance of $C_7$ is more questionable, as it just shifts the total cost of each design by a constant value. It may make this cost breakdown more "realistic" in a practical sense, but does it impact how the authors' proposed design approach would be utilized? Were these fixed costs larger or equal in size compared to the other terms, one might conclude that differences between the topologies were negligible and therefore not worth (or to a lesser extent worth) pursuing, but this is not the case here. In any case, this term needs more justification by the authors.

- What is the sensitivity of the optimal designs to each term in the cost function? If possible, what would the optimal designs look like if some/certain terms were neglected? If that is too comprehensive, what is the contribution of each term in the cost function to the gradient of the objective function? Especially those elements of the gradient corresponding to the most design driving variables. Evaluate this at, e.g., the initial design, an intermediate design and close to the optimal design. Comparing terms that are more highly dependent on tube dimensions to the ones that depend more strongly on topology, does the inclusion of the latter terms change the direction of the gradient or does it merely reinforce the steps the algorithm would otherwise take?

Given that $C_1$, $C_2$ and $C_3$ (and hence also $C_5$) all depend in some way on effective tube dimensions (changing mass, weld volume, outer areas), does the different weighting of

the design variables induced by the inclusion of all three terms in the cost function have a significant impact on the optimal design? In other words, since several of the cost terms are in a sense "proportional" (partially or otherwise) to the total mass, how much is changed by the inclusion all of these terms (rather than just the mass)? Clearly, these terms contribute significantly to the actual cost. However, given the correlation with mass, do they have a large effect on the solution of the optimization problem?

Similarly, since many of the design variables controlling topology also enter into the cost terms related to mass, does the inclusion of cost terms entirely related to topology have a significant impact on how these variables are changed by the optimization process? For example, it seems like the costs related to the transition piece is almost completely determined by the number of legs, since the values of $R_{foot}$ and $\xi$ are the same (or almost the same) for all design with the same number of legs. One then wonders if the values of $R_{foot}$ and $\xi$ are actually determined (or at least significantly affected) by the inclusion of the transition piece cost term, or if similar behavior would be seen without this term. If so (and if this was seen to be a more general result also outside the scope of the present study), this would mean that the cost of the transition piece would not need to be included in the continuous optimization problem, but only added in along with the installation cost as an additional cost related to the number of legs.

Shedding some light on these issues would considerably strengthen the proposed cost-model methodology compared to previous studies using just mass optimization.

- On page 16, lines 6-7, you write "The number of iterates may be decreased, when using finite differences of the objective function to obtain gradients ..." What exactly do the authors mean here? Having precise analytical gradients of the objective function would generally tend to improve the behavior of optimization routines, since this is less prone to numerical error.

6. Benefits and limitations of the approach:

- On page 17, lines 24-26, you write "... these approaches assume that the structural

topology is always optimal, even in case of significant variations in tube dimensions. However, when combined with the approach presented in this work, state-of-the-art tube dimensioning may be much more powerful." If possible, please comment on how the inclusion of variables related to design topology changes how the structure is optimized compared to pure tube size optimization. I.e. to what extent is the reduction (or change overall) of tube size "replaced" by changes to topology?

**Technical corrections:**

- Abstract, page 1, line 6: "The objective function is replaced by a sum term..." For clarity, replace by "The conventional mass objective function is replaced by a sum of terms..." or equivalent.

- Abstract, page 1, line 8: "... numerical cost ..." Since "cost" was already used with a different meaning, replace by a different word to avoid confusion.

- Section 1, page 2, line 20: "... legs or bays are rather interesting than the exact dimensions". Replace "rather interesting" by "more critical" or similar.

- Section 1, page 3, line 2: "More realistic load sets..." Use "comprehensive" instead of "realistic".

- Section 2, page 3, line 31: "... are not impacted by design variables." Change to "... are not impacted by the selected design variables."

- Section 2, page 4, lines 1-4: I would recommend combining these two list entries into a single entry, since they are closely related.

- Section 3.2, page 7, line 13: Define the subscript index $m$ in the text.

- Section 3.3, page 9, lines 13-14: Replace "A number of 128..." with "128 design load cases ..."

- Section 4, page 10, line 19: "... when the number of constraints and design variables are in the same dimension." Replace "dimension" with "order of magnitude".

- Section 5.2, page 12, line 1: "... the soil layup of the Offshore Code ..." Replace "layup" with "conditions" and/or "model" depending on what the intended meaning is.

- Section 5.4, page 16, line 6: "The number of iterates may ..." Replace "iterates" by "iterations".

- Section 5.4, page 16, lines 7-8: "... but is not vital in this dimension of computational expenses." Replace "in this dimension" with "at this level".
* * *

---

## Author Comment (AC1) · 5 Nov 2018

Dear reviewer,

I thank you for your highly valuable and comprehensive comments on the paper. Probably, it took much time to work on this review, which is really appreciated. Most of the comments have been incorporated directly in a revised version, which is attached to this response. The particular responses are given in the following.

*On page 1, lines 12-13, you write: "The approach shows reasonable and promising results, ..." Even for the abstract, this is a little vague. You should say something about how the method seems to succeed in finding global minima with reasonable*

*computational effort and how the method allows for a systematic way of comparing cost and structural performance for optimal designs with different topologies.*
I agree, this was too vague. In the revised version, the abstract tells more details at this point.

*On page 2 you have a fairly comprehensive literature review. You might also want to consider adding a note about Sandal et al (2018, Marine Structures) where the effect of varying leg distance (functionally equivalent to what you call leg radius) was studied to a certain extent.*
When the state of the art was surveyed, this paper was not published yet. Of course, I added it to the literature review. In addition, I added the reference Oest et al. (2018) for the same reason.

*On page 4, lines 2-4, you write about the limited effect of changes to eigenfrequencies for the design. This would likely be a larger (if not necessarily critical) issue if the soil and foundation were modeled in more detail than what you have done in this study. Though you correctly note that any design close to resonance would fail the fatigue check, there are practical reasons (related to the numerical behavior of the optimization process) that can make it beneficial to include explicit constraints on the eigenfrequency rather than relying exclusively on the fatigue constraints.*
I agree, the reason why it is not required to incorporate eigenfrequency constraints is that the foundation is maintained in any case. I extended the sentence to clarify this.

*On page 7, line 10, you define c7. This is constant for all designs and hence has no effect on the optimization process. While it may be instructive to note the existence of such additional costs in practice, for the optimization problem being solved such a term is irrelevant. Hence, you should at the very least state whether this term is part of the implemented computations or if it is merely added at the end of the optimization.*

For the optimization process, the constant terms are excluded from the objective function, but I believe that the results are easier to compare, when they show total costs including the constant terms. I added this information in section 5, subsection 5.4.

*On page 9, lines 3-4, you describe a "space-filling sampling of the input space." As far as I can see, the details of this sampling is not given either here or in the previous paper about the surrogate modeling. It would be instructive to have some more details here. At the very least, specifically what kind of sampling method are you using?*
It is a Latin hypercube sampling, I added this information,. I don't think it is necessary to go deeper into details here, as the focus is not on the surrogate model.

*On page 9, lines 13-14, you write "A number of 128 turned out to be a good compromise between accuracy and numerical effort." The details of this was given in previous work, but for the sake of the reader of the present work it would be instructive to give a quick summary of what level of accuracy 128 load cases represents.*
I agree. The description was extended.

*On page 9, lines 16-17, you write "The output value $h_{FLS}$ is the most critical fatigue damage among all damage values of the entire jacket..." Considering the results you present later, this does not seem to have been a problem here, but note that using such a constraint (the maximum of a discrete set) could lead to discontinuities that affect your gradients (sensitivities) adversely."*

and

*On page 9, lines 28-29, you write "... evaluated with respect to the extreme load of the member, where the highest utilization ratio occurs." Here you have the same potential issue as with the fatigue constraint.*
I agree to both comments. Prior to this study, I experimented with the constraints. I know that it is better to incorporate each fatigue damage as a constraint, which is not a big issue and from the numerical point of view the better solution. However, the idea of the surrogate model was to provide a solution for simple and quick fatigue evaluations.

*On page 9, line 27, you write "Extreme load parameters are derived by the block maximum method according to Agarwal and Manuel (2010)." This needs a little more detail. Does this include a statistical extrapolation to a 50-year (or n-year) return value?*
Yes, it does. I added a short description, what "block maximum" means and how the load extrapolation is performed.

*On page 12, lines 18-21, you summarize the different fixed integer design variable combinations you study. Note that term $c_6$ in your previously defined cost function is constant for each of these combinations and hence has no (computational) effect on the individual optimization problems (merely adds a constant term to the solution).*
Same as above. The information was added to section 5, subsection 5.4.

*On page 12, lines 27-33 [also on page 16, lines 3-6], you describe the behavior of your optimization routines. It would be instructive to plot an example of the convergence. For example cost (and maybe feasibility) vs number of iterations. How many function evaluations is typically involved per iteration? The number of function evaluations (in terms of both objective and constraints) is a more direct measure of computational speed/effort than the number of iterations (and is more easily generalized to different machines). It also says something about how the algorithm is behaving.*
I've added a figure (Fig. 2) showing the convergence behavior of both optimization methods and all six subproblems with the OC4 jacket as starting solution. Subsection 5.4 has also been extended.
*On pages 13-14, you discuss the properties of the optimal designs. What do the initial designs look like (especially in terms of cost and feasibility with respect to code checks)? How are the optimal designs compared to a "typical" initial design, maybe compared to the OC4? Is the most optimal design topology $(N_L, N_X) = (3,3)$ also the one with the most improvement compared to "initial" designs? How is this for the various other topologies? Obviously, the results here are only meant to illustrate the method, so the specifics of the optimal designs are not so important. However, it would give more insight into the effect of your cost model and your chosen design variables if you explain in more detail what kind of optimal designs your methodology tends to produce, with more clear reference to initial designs. What would you say are the design driving variables?*

I agree that these questions are important. From my perspective, Fig.2 also answers most of these questions. Moreover, I addressed these points in the text, subsection 5.4.

*On page 14, line 5, you write "Altogether, this is meaningful and not far off from structural designs that are known from practical applications." Do the authors mean that the designs are close to that seen in practical applications or that the costs are? Please elaborate a bit on this statement.*

Both the designs (in terms of number of legs, which shows a trend to three-legged jackets) and the costs are close to practical applications. I've modified the sentence to make it clearer.

*On page 16, Figure 3, you show a cost breakdown of the optimal designs. Since $C_1$ and $C_5$ both depend on the mass, note that the mass is associated with the largest proportion of the cost, but this is not immediately obvious from the figure. From a practical point of view, these two points indeed represent different aspects of the production and installation process, but since both these terms directly contribute to lighter designs, their effects in an optimization context are the same (and in the figure,*

*these two terms are clearly just scaled versions of each other). While $C_6$ does not impact the optimization directly, its presence in the cost breakdown is justified by how it clearly shows where a significant portion of the difference in total cost between 3- and 4-legged designs come from. The importance of $C_7$ is more questionable, as it just shifts the total cost of each design by a constant value. It may make this cost breakdown more "realistic" in a practical sense, but does it impact how the authors' proposed design approach would be utilized? Were these fixed costs larger or equal in size compared to the other terms, one might conclude that differences between the topologies were negligible and therefore not worth (or to a lesser extent worth) pursuing, but this is not the case here. In any case, this term needs more justification by the authors.*

This is a very good point. I compared the cost function to a pure mass-dependent one at the end of subsection 5.4. The result is that one obtains similar designs (I've added Table 3 with new results), when only the mass is considered as objective. The reason is obvious (as written in the comment). $C_1$ and $C_5$ are proportional to mass, $C_6$ and $C_7$ do not impact the optimization problem, $C_2$ and $C_3$ are in some way proportional to tube dimensions. The only remaining cost term, $C_4$, is affected by topological variables. However, these variables have a greater impact on structural resistance than on costs.

*What is the sensitivity of the optimal designs to each term in the cost function? If possible, what would the optimal designs look like if some/certain terms were neglected? If that is too comprehensive, what is the contribution of each term in the cost function to the gradient of the objective function? Especially those elements of the gradient corresponding to the most design driving variables. Evaluate this at, e.g., the initial design, an intermediate design and close to the optimal design. Comparing terms that are more highly dependent on tube dimensions to the ones that depend more strongly on topology, does the inclusion of the latter terms change the direction of the gradient or does it merely reinforce the steps the algorithm would otherwise take?*

*Given that $C_1$, $C_2$ and $C_3$ (and hence also $C_5$) all depend in some way on effective tube dimensions (changing mass, weld volume, outer areas), does the different weighting of the design variables induced by the inclusion of all three terms in the cost function have a significant impact on the optimal design? In other words, since several of the cost terms are in a sense "proportional" (partially or otherwise) to the total mass, how much is changed by the inclusion all of these terms (rather than just the mass)? Clearly, these terms contribute significantly to the actual cost. However, given the correlation with mass, do they have a large effect on the solution of the optimization problem?*

*Similarly, since many of the design variables controlling topology also enter into the cost terms related to mass, does the inclusion of cost terms entirely related to topology have a significant impact on how these variables are changed by the optimization process? For example, it seems like the costs related to the transition piece is almost completely determined by the number of legs, since the values of Rfoot and $\xi$ are the same (or almost the same) for all design with the same number of legs. One then wonders if the values of Rfoot and $\xi$ are actually determined (or at least significantly affected) by the inclusion of the transition piece cost term, or if similar behavior would be seen without this term. If so (and if this was seen to be a more general result also outside the scope of the present study), this would mean that the cost of the transition piece would not need to be included in the continuous optimization problem, but only added in along with the installation cost as an additional cost related to the number of legs.*

*Shedding some light on these issues would considerably strengthen the proposed costmodel methodology compared to previous studies using just mass optimization.*
As proposed, I computed the sensitivities of each cost function term that is impacted by the design variables, shown in Fig. 4 at the initial design, an intermediate design, and

the optimal design. The results are discussed in subsection 5.4. Also, the evaluation of the mass-dependent approach (described above) sheds light in the cost model methodology. Using the comprehensive cost model does not yield completely different results. However, from my point of view, it is a good result that mass-dependent approaches are actually more accurate than expected.

*On page 16, lines 6-7, you write "The number of iterates may be decreased, when using finite differences of the objective function to obtain gradients ..." What exactly do the authors mean here? Having precise analytical gradients of the objective function would generally tend to improve the behavior of optimization routines, since this is less prone to numerical error."*
I extended the sentence to make it clearer.

*On page 17, lines 24-26, you write "... these approaches assume that the structural topology is always optimal, even in case of significant variations in tube dimensions. However, when combined with the approach presented in this work, state-of-the-art tube dimensioning may be much more powerful." If possible, please comment on how the inclusion of variables related to design topology changes how the structure is optimized compared to pure tube size optimization. I.e. to what extent is the reduction (or change overall) of tube size "replaced" by changes to topology?*
It is not easy to quantify this (for instance, because the OC4 jacket is just a derivation of the UpWind jacket, which is a detailed design), but it is replaced to some extent when comparing the topologies. I modified this paragraph to address this point.

*Technical corrections*
I adopted all suggestions. Thank you for these corrections.

I hope that the revision and my comments are satisfactorily. I would appreciate

a recommendation of the reviewer for publication in Wind Energy Science.

Best regards,
Jan Häfele
(on behalf of all authors)

Please also note the supplement to this comment:
https://www.wind-energ-sci-discuss.net/wes-2018-58/wes-2018-58-AC1-
supplement.pdf

**Supplement:**

[revised manuscript text omitted]

---

## Referee Comment (RC2) · Anonymous Referee #2 · 12 Nov 2018

**Review of "A comparison study on jacket subsctructuresfor offshore wind turbines based on optimization", manuscript no.: wes-2018-58, by J. Häfele, C.G. Gebhardt, and R. Rolfes**

The manuscript addresses the structural optimization problem of jacket substructures for offshore wind turbines. It takes a broader approach than most papers within the field by considering topological variations in the optimization, extending to e.g. number of bays and amount of legs. Additionally, the work includes a more detailed cost function than most optimization papers.

Overall, the topic is worth investigating and important for further improving the design of offshore jacket substructures for wind turbines. The topic is within the scope of Wind Energy Science and the article advances current state-of-the-art. However, I do have some suggestions and comments that should be addressed.

Generally, the modeling section should be more descriptive and the language in especially the result section should be revised.

The choices behind the cost-model should be described in much more detail. Also, the limitations of the cost model should be reflected upon. For instance, local content is a large factor in the current market. Thus, designs can to a certain extent be driven by the locally available production facilities. To name an example, this can affect the number of bays due to crane facilities or painting facilities. While it is fair not to include all the aspects, more important factors should at least be discussed.

The cheapest structures appear to be the simplest structure, i.e. the fewest bays. This is intuitive, as it is well-known that welding and potentially grinding of jacket structures is very expensive. Thus, the motivation for implementing this framework is lacking, as you get the expected result, as you also mention in the paper. However, if sensitivities to different terms of the cost function were presented, much more insight into the design drivers would be given, and this would add significant value to the paper. E.g., how much would you need to lower the production cost, for instance by robot-welding of X-braces, before we get a different optimal design?

Lastly, since the cost model aims to replace the more used 'overall mass' model, the overall weight of each of the optimized structures should clearly be stated in the result section. The cheapest four-legged jacket is the lightest four-legged jacket. How about the three-legged? This information is lacking. It would have been very convenient to see a minimization of mass optimization compared to the presented results.

**Specific comments:**

**Page 1**

Line 20 — You mention that structural optimization is paramount (I do agree, at least in absence of experts) because it provides cost savings "with low effort". Low effort in execution, yes, but not necessarily in implementation of the method. More focus should be on how easy or difficult it is to implement the proposed optimization method.

| Line 21-22 | For clarity, I suggest that you directly mention what is meant by 'intermediate water depths'. |
|---|---|

**Page 2**

| Line 3-4 | It is true that thousands of simulations are required for verification, but it should be clarified, that it is not needed during conceptual design phases with or without optimization methods. |
|---|---|
| Line 19-20 | You do not mention decision by design 'experts' until page 3, but number of bays and legs are normally correctly decided by experienced designers. Consider restructuring/rephrasing. |

**Page 3**

| Line 30-31 | This is an assumption. Pile design can be affected by the design of the substructure. |
|---|---|

**Page 4**

| Line 12 | You should mention why the cost function is scaled with log10. If you experienced numerical difficulties without the logarithmic scaling, this should also be mentioned. |
|---|---|
| Line 21-22 | 'The problem incorporates no nonlinear equality constraints'. This sentence can be removed. This is clearly stated in equation 3. |

**Page 5**

| Figure 1 | The last sentence in the figure text lacks a 'respectively' or should be rephrased. |
|---|---|
| Section 3.1 | It is fair to reduce the design space by always having a mudbrace, but real jacket structures do not always have this. The impact on both the structural response and on the manufacturability/costs of having a mudbrace or not should be mentioned. |

**Page 6**

| Equation (5) | You should mention that the actual weights are presented in section 5.3 or the weights should be listed here. |
|---|---|

**Page 7**

| Section 3.2 | Generally, the limitations and assumptions of the equations should be made much more visible. While this part is a large step forward in defining the optimization problem as compared with most previous work, the cost function is still quite simplified. |
|---|---|
| Equation (10) | You assume that the transport cost is directly dependent on the mass. This is a very large simplification, and effectively makes the additional constraint obsolete at is just an additional factor on C1.
I fully realize that it may be too complicated to incorporate many of the governing factors, e.g. crane and vessel availability. However, e.g. deck space occupied by a three- or four-legged jacket is very different, and this can have a significant impact on the transportation and installation costs. |
| Equation (11) + (12) | I think that there should be a difference in the cost function for an optimization problem, and the actual costs. There is no need to add fixed costs to the optimization problem. |

**Page 9**

Section 3.3.1     It should be clearer that the Efthymiou SCF's are just one way of determining the SCF's, and they are well-known to be quite unprecise. People that are unfamiliar with fatigue design of offshore structures may believe that this is the standard approach, which is most often not the case.

**Page 10**

Section 4.1 + 4.2     Not enough details are given. E.g. what are 'appropriate settings' for fmincon?

**Page 14**

Table 1     Can you include the overall mass of the jacket in this table? This is important to compare the results from standard 'mass minimization' to your 'cost minimization'.

It looks like the cheapest jackets are also the lightest jackets.  At least, the cheapest 4-legged jacket is the lightest jacket. It could be very interesting to see if the cheapest 3-legged jacket is also the lightest 3-legged jacket.

---

## Author Comment (AC2) · 21 Nov 2018

Dear reviewer,

I appreciate your effort for this comprehensive review. I did my best to address all comments and revised the manuscript (see attachment) based on the first revision, which addressed the first reviewer's comments (RC1). The particular responses are given in the following.

*Generally, the modeling section should be more descriptive and the language in especially the result section should be revised.*

I extended the modeling section, especially the cost modeling part. In addition, more detailed descriptions of modeling (also the structural code checks obtained by surrogate modeling) can be found in another work, which is referenced in the introduction of section 3. The shortcomings of the cost model are discussed in more detail in the benefits and limitations, section 6. I also revised the language.

*The choices behind the cost-model should be described in much more detail. Also, the limitations of the cost model should be reflected upon. For instance, local content is a large factor in the current market. Thus, designs can to a certain extent be driven by the locally available production facilities. To name an example, this can affect the number of bays due to crane facilities or painting facilities. While it is fair not to include all the aspects, more important factors should at least be discussed.*

See the previous point. Further explanations and discussions are given in sections 3.2 and 6.

*The cheapest structures appear to be the simplest structure, i.e. the fewest bays. This is intuitive, as it is wellknown that welding and potentially grinding of jacket structures is very expensive. Thus, the motivation for implementing this framework is lacking, as you get the expected result, as you also mention in the paper. However, if sensitivities to different terms of the cost function were presented, much more insight into the design drivers would be given, and this would add significant value to the paper. E.g., how much would you need to lower the production cost, for instance by robot-welding of X-braces, before we get a different optimal design?*

This is correct. A similar point was annotated by the first reviewer (RC1) and I performed a little sensitivity analysis, where the variation of each cost function term was studied with respect to variations in design parameters.

*Lastly, since the cost model aims to replace the more used 'overall mass' model, the overall weight of each of the optimized structures should clearly be stated in the result section. The cheapest four-legged jacket is the lightest four-legged jacket. How about the three-legged? This information is lacking. It would have been very convenient to see a minimization of mass optimization compared to the presented results.*

This is also correct. I already performed an optimization loop using a mass-dependent approach and compared the results in the first revision.

*You mention that structural optimization is paramount (I do agree, at least in absence of experts) because it provides cost savings "with low effort". Low effort in execution, yes, but not necessarily in implementation of the method. More focus should be on how easy or difficult it is to implement the proposed optimization method.*

I thought about how to address this comment appropriately. What you say, is absolutely correct. However, it is meant that cost savings can be reached just by improving the design process. Of course, it requires effort in implementing the method, but not much economical effort. Therefore, I decided to write "with low economical effort" and added the remark to the benefits and limitations section (sect. 6).

*For clarity, I suggest that you directly mention what is meant by 'intermediate water depths'*

I clarified this.

*It is true that thousands of simulations are required for verification, but it should be clarified, that it is not needed during conceptual design phases with or without optimization methods.*

This was clarified in a footnote.

*You do not mention decision by design 'experts' until page 3, but number of bays and legs are normally correctly decided by experienced designers. Consider restructuring/rephrasing.*

I rephrased the sentence.

*This is an assumption. Pile design can be affected by the design of the substructure.*

I added ". . .in this approach" to make clear that this is an assumption of this work.

*You should mention why the cost function is scaled with log10. If you experienced numerical difficulties without the logarithmic scaling, this should also be mentioned.*

I added this information.

*'The problem incorporates no nonlinear equality constraints'. This sentence can be removed. This is clearly stated in equation 3.*

The sentence was removed.

*The last sentence in the figure text lacks a 'respectively' or should be rephrased.*

I added "respectively" at the end of the sentence.

*It is fair to reduce the design space by always having a mudbrace, but real jacket structures do not always have this. The impact on both the structural response and on the manufacturability/costs of having a mudbrace or not should be mentioned.*

I fixed the mud brace flag, because it is not a continuous parameter. This is stated now.

*You should mention that the actual weights are presented in section 5.3 or the weights should be listed here.*

My intention was to split the descriptions of models and parameters. I put a reference to the unit cost values in a footnote.

*Generally, the limitations and assumptions of the equations should be made much more visible. While this part is a large step forward in defining the optimization problem as compared with most previous work, the cost function is still quite simplified.*

See my first particular response.

*You assume that the transport cost is directly dependent on the mass. This is a very large simplification, and effectively makes the additional constraint obsolete at is just an additional factor on C1. I fully realize that it may be too complicated to incorporate many of the governing factors, e.g. crane and vessel availability. However, e.g. deck*

*space occupied by a three- or four-legged jacket is very different, and this can have a significant impact on the transportation and installation costs.*

I'm aware that this is a large simplification. My idea was to consider transport costs to some extent by a simple approach, because it is difficult for me to get realistic cost values here. I talked with people from industry and decided to select the mass as governing factor, as the mass is on the one hand at least partially influencing the transport costs and on the other hand partially related to other measures of the jacket affecting the transport costs (like, for example, deck space occupied by the structure). From the optimization perspective, $C_5$ is proportional to $C_1$ and therefore not necessary, I agree (see my comments to the first reviewer). My intention was to separate material and transport costs, which leads to a more realistic cost breakdown in the results. Someone, who has a better model for installation costs may, however, replace $C_5$ by a more detailed term.

*I think that there should be a difference in the cost function for an optimization problem, and the actual costs. There is no need to add fixed costs to the optimization problem.*

For the solution of the optimization problem, fixed terms are excluded from the objective function. This information was missing, see my comments to the first reviewer.

*It should be clearer that the Efthymiou SCF's are just one way of determining the SCF's, and they are well-known to be quite unprecise. People that are unfamiliar with fatigue design of offshore structures may believe that this is the standard approach, which is most often not the case.*

I clarified this.

*Not enough details are given. E.g. what are 'appropriate settings' for fmincon?*

I removed the term "with appropriate settings" from this sentence. Instead, I improved the beginning of section 5.4, where I describe some settings of the optimization methods. Additionally, the convergence behavior is shown and discussed in the revised

manuscript. I believe that this strengthens the (mathematical) optimization aspect.

*Can you include the overall mass of the jacket in this table? This is important to compare the results from standard 'mass minimization' to your 'cost minimization'. It looks like the cheapest jackets are also the lightest jackets. At least, the cheapest 4-legged jacket is the lightest jacket. It could be very interesting to see if the cheapest 3-legged jacket is also the lightest 3-legged jacket.*

The row with overall masses was added to the table. As $C_1$ (and $C_5$) are proportional to mass, the overall masses were given in Fig. 3. However, I agree that this was not very obvious. I discussed it in the text in more detail.

I hope that the revision and my comments are to your satisfaction and would appreciate a recommendation for publishing the revised manuscript in Wind Energy Science.

Best regards,
Jan Häfele
(on behalf of all authors)

Please also note the supplement to this comment:
https://www.wind-energ-sci-discuss.net/wes-2018-58/wes-2018-58-AC2-supplement.pdf

**Supplement:**

[revised manuscript text omitted]

---

## Author Comment (AC3) · 21 Nov 2018

Immediately after submitting my comment I realized that the subject was wrong. Of course, the subject should be "Response to referee comment (RC2)"

I'm sorry, if I confused anyone!

Best regards,
Jan Häfele

---

## Referee Report (RR1)

**General comments**

The new additions to the paper have strengthened the value of the results and made them more useful in general, allowing for some additional conclusions to be drawn that might previously have been only implied or guessed. Especially the comparison with a mass-only cost function gives the reader a good idea about when the additional complexity of the proposed approach is needed. With a few minor additions, this work will be ready for publication.

**Specific comments**

Section 3.3.1, page 10, lines 10-12: The newly added explanation of the size of the reduced load set would be even more clear if the expected uncertainty was quantified. I.e. how many % is the expected error in the computed fatigue damage?

Section 5.4, page 18, Figure 4: To more clearly show the impact of each variable on the total cost (objective function), you should add data in each plot for the variation of the total cost, $\Delta C_{total}$.

**Technical corrections**

- Abstract, page 1, line 6: The suggestion for a change to this sentence was probably phrased badly in the previous review. For clarity, change "... a sum of terms." to " ... a sum of various terms related to the cost of the structure."
- Section 3.2, page 8, line 21: "... a measure related to the actual costs." Change to " ... a measure of the actual costs."
- Section 3.2, page 9, line 2: "The factor mass ..." Change to "The mass-dependence ..."